# Scalable Multi-Task Low-Rank Model Adaptation

**Zichen Tian    Antoine Ledent    Qianru Sun**
Singapore Management University
`zichen.tian.2023@phdcs.smu.edu.sg, {aledent, qianrusun}@smu.edu.sg`

## Abstract

Scaling multi-task low-rank adaptation (LoRA) to a large number of tasks induces catastrophic performance degradation, such as an accuracy drop from 88.2% to 2.0% on DOTA when scaling from 5 to 15 tasks. This failure is due to parameter and representation misalignment. We find that existing solutions, like regularization and dynamic routing, fail at scale because they are constrained by a fundamental trade-off: strengthening regularization to reduce inter-task conflict inadvertently suppresses the essential feature discrimination required for effective routing. In this work, we identify two root causes for this trade-off. First, uniform regularization disrupts inter-task knowledge sharing: shared underlying knowledge concentrates in high-SV components (89% alignment on Flanv2→BBH). Uniform regularization forces high-SV components to update in orthogonal directions, directly disrupting the shared knowledge. Second, Conflict Amplification: Applying LoRA at the component-level (e.g., $W_q, W_v$) amplifies gradient conflicts; we show block-level adaptation reduces this conflict by 76% with only 50% parameters. Based on these insights, we propose mtLoRA, a scalable solution with three novel designs: 1) Spectral-Aware Regularization to selectively orthogonalize low-SV components while preserving high-SV shared knowledge, 2) Block-Level Adaptation to mitigate conflict amplification and largely improve parameter efficiency, and 3) Fine-Grained Routing using dimension-specific weights for superior expressive power. On four large-scale (15-25 tasks) vision (DOTA and iNat2018) and NLP (Dolly-15k and BBH) benchmarks, mtLoRA achieves 91.7%, 81.5%, 44.5% and 38.5% accuracy on DOTA, iNat2018, Dolly-15k and BBH respectively, outperforming the state-of-the-art by 2.3% on average while using 47% fewer parameters and 24% less training time. Code is available at `https://github.com/doem97/ICLR26_mtLoRA`.

## 1 Introduction

Low-Rank Adaptation (LoRA) (Hu et al., 2021) has emerged as the *de-facto* standard of Parameter-Efficient Fine-Tuning (PEFT) for pre-trained Visual Transformer (ViT) models, thanks to its minimal trainable parameters, zero inference latency overhead, and modular deployment (He et al., 2022; Zhang et al., 2023; Dettmers et al., 2023; Han et al., 2024; Ge et al., 2025; Tian et al., 2025; Zhu et al., 2025; 2024). Although LoRA achieves remarkable performance in single-task adaptation (Zhang et al., 2023; Liu et al., 2024; Tian et al., 2024b), real-world applications usually need scalable multi-task low-rank adaptation, *i.e.*, using multiple task-specific LoRA modules (on top of one backbone model) to handle a large number of tasks (15-25+) simultaneously (Stoica et al., 2025; Wu et al., 2024a; Ma et al., 2018). For instance, language models need to process multiple tasks (*e.g.*, mathematical reasoning, legal analysis, and ethical questions) concurrently (Hendrycks et al., 2020; Zhao et al., 2025), and vision models need to adapt across multiple spectrums (*e.g.*, optical and radar imagery) (Tian et al., 2024b). Training large foundation models from scratch for domain-specific applications presents fundamental challenges, particularly in domains with limited data availability and severe data imbalance issues (Wang et al., 2025), further motivating parameter-efficient multi-task approaches. However, multi-task low-rank adaptation suffers from catastrophic performance degradation as the number of tasks scales up (Tian et al., 2024a; Wu et al., 2024a; Stoica et al., 2025).

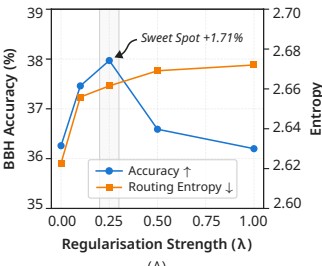 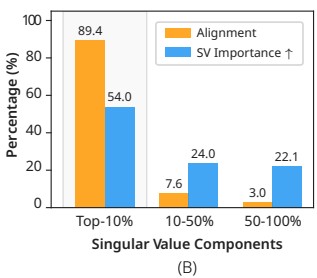 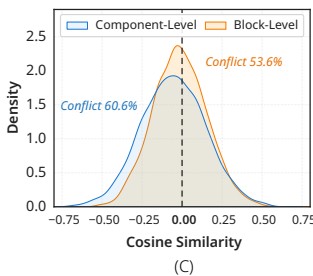

(A)  (B)  (C)

Figure 1: **Motivating observations for our three novel designs. (A) Orthogonal regularization introduces a trade-off between conflict reduction and routing uncertainty.** Specifically, through orthogonal regularization, the model accuracy (blue curve) peaks at $\lambda = 0.25$ (+1.7%) but degrades at $\lambda = 1.0$ (-1.8%), due to increased routing uncertainty (i.e., Routing Entropy in orange curve). **(B) Shared knowledge concentrates in high-SV components.** Specifically, high-SV (top-20%, highlighted) shows 89% inter-task alignment and encodes 54% of total singular values, while low-SV (50-100%) shows only 3% alignment with 22% of singular values (detailed in Sec. 4.1). This motivates spectral-aware regularization: preserve high-SV shared knowledge, only orthogonalize low-SV components. **(C) Block-level LoRA adaptation reduces gradient conflicts.** Specifically, block-level adaptation achieves higher gradient alignment between tasks (measured by cosine similarity, $-0.013_{\pm 0.169}$) as compared to component-level adaptation ($-0.054_{\pm 0.201}$), accompanied by a +2.1% accuracy improvement (91.2% vs. 89.0% in Table 4). See Sec. 4.3 for detailed experimental setups.

The core challenges are two kinds of misalignment: parameter misalignment and representation misalignment (Stoica et al., 2025; Han et al., 2024). Specifically, **parameter misalignment** means different LoRA modules have conflicting weight updates (*i.e.*, gradient of weights update in opposing directions). To address this, existing methods use regularizations to enforce orthogonality across LoRA parameters (Ilharco et al., 2022; Yadav et al., 2023; Yu et al., 2024a). Another challenge is **representation misalignment**, meaning that LoRA modules' output features are divergent (Stoica et al., 2025). Existing solutions use dynamic routing to weigh LoRA's output features, *i.e.*, by sparse gating (*e.g.*, select top-K activated LoRA) or soft routing (*i.e.*, weighted combining all LoRA modules) (Wu et al., 2024a; Wei et al., 2025; Tian et al., 2024a). However, these methods fail to scale to large numbers of tasks. An intuitive solution is to combine both approaches to leverage their complementary strengths. We find that, while this combination improves performance, it quickly reaches a Pareto frontier: stronger regularization reduces gradient conflict but impairs routing effectiveness. Specifically, as shown in Figure 1(A), when training LoRA modules with dynamic routing on a multi-task NLP scenario (trained on 16 experts Flan-v2 and evaluated on BBH), increasing the regularization strength $\lambda$ initially improves the accuracy to 38.0% (+1.7% at $\lambda = 0.25$). However, further strengthening regularization increases the routing uncertainty (i.e., routing entropy increases from 2.6 to 2.7) and causes accuracy to drop by 1.8%. This trade-off fundamentally limits scalability.

This raises a key question: **why does this trade-off exist?** We identify two root causes stemming from how LoRA modules are treated and placed, respectively.

**First, uniform regularization disrupts knowledge sharing across tasks.** The key of multi-task learning is to share underlying knowledge (*i.e.*, transferring inductive bias in (Caruana, 1997)) across tasks. However, uniform regularization disrupts this. We quantify the shared knowledge across tasks in Figure 1(B). Results show that the shared knowledge concentrates in high-SV components, *i.e.*, high-SV (top-20%) contains 89% inter-task alignment, meanwhile encodes 54% of total singular values; while low-SV (50-100%) contains only 3% alignment with 22% of singular values (see Sec 4.1). Uniform regularization treats all spectral components equally, forcing orthogonality on high-SV and pushing them to update in different directions, which directly corrupts the knowledge sharing. This motivates our spectral-aware regularization, *i.e.*, orthogonalize low-SV noise, while preserving high-SV shared knowledge. **Second, applying LoRA to component-level matrices amplifies gradient conflicts.** When multiple LoRA modules adapt individual weight matrices (*e.g.*, $W_q$, $W_v$) for different tasks, their gradients exhibit stronger misalignment. Figure 1(C) validates this: component-level adaptation yields an average gradient cosine similarity of $-0.054_{\pm 0.201}$ between task pairs, indicating substantial conflict. In contrast, block-level adaptation (applying LoRA to

entire attention and FFN blocks) reduces this to $-0.013_{\pm0.169}$ (76% reduced conflict). This improved gradient alignment leads to +2.2% accuracy gain (91.2% vs. 89.0%, Table 4).

Given these insights, we propose $\mathtt{mtLoRA}$–a novel method designed for scalable multi-task low-rank adaptation. We introduce three key designs: 1) spectral-aware regularization, 2) fine-grained routing, and 3) block-level adaptation. **First, we design spectral-aware regularization.** Our approach applies strong orthogonalization to low-SV components (empirically, identified as "noise") while preserving high-SV components (the "signal"). We achieve this through a weighting function $w(\sigma) = \exp(-\sigma/\bar{\sigma})$, where $\sigma$ is the singular value and $\bar{\sigma}$ is the average. For noisy low-SV components (*i.e.*, $\sigma \ll \bar{\sigma}$), the weight $w(\sigma)$ approaches 1, enforcing strong orthogonality. For discriminative high-SV components (encoding shared underlying knowledge, *i.e.*, $\sigma \gg \bar{\sigma}$), the weight $w(\sigma)$ approaches 0, preserving inter-task knowledge sharing. **Second, we propose fine-grained routing.** Unlike standard routing that assigns each LoRA a scalar weight (forcing a uniform combination across all dimensions), we learn a router network to produce a vector $\Pi_i \in \mathbb{R}^g$ to weigh each LoRA, where $g$ is the number of groups. Each group contains $d/g$ dimensions. This addresses the observed heterogeneous conflict pattern, allowing different feature subspaces to use different combinations of task experts. For example, for a complex prompt, a "creativity" subspace can assign a high weight to a brainstorming LoRA, while a "factual" subspace can simultaneously assign a high weight to a QA LoRA. Our fine-grained routing breaks the constraint of uniform combination and is stabilized by a load-balancing loss to prevent routing collapse. **Third, we propose block-level adaptation.** Rather than adapting individual matrices within a block, we apply the combined LoRA update as a parallel path that bypasses the block's internal computations. Consistent with Pre-LN architectures, for a block $F$ and its LayerNorm LN, our adapted output is $h_{out} = h_{in} + F(\text{LN}(h_{in})) + \Delta(\text{LN}(h_{in}))$. The LoRA adapter $\Delta$ operates on the same normalized input as $F$, mitigating multiplicative gradient conflicts and ensuring architectural consistency.

We validate our $\mathtt{mtLoRA}$ on four large-scale multi-task benchmarks spanning both vision and language domains: DOTA (15 tasks), iNat2018 (25 tasks), Dolly-15k (16 tasks), and BBH (27 reasoning tasks). We make three key findings. **1) $\mathtt{mtLoRA}$ is more scalable.** Existing methods face severe collapse when task number increases, For example, naive averaging leads to catastrophic degradation: 88.2%→2.0% on DOTA (5→15 tasks) and 87.0%→0.3% on iNat2018 (1→100 tasks in Supplementary Sec C.1). Our $\mathtt{mtLoRA}$ mitigates this collapse, achieving 64.0% average accuracy across all four benchmarks, outperforming HydraLoRA by 2.3% on average (Table 5). Ablation studies confirm **2) all three key designs contribute significantly.** Specifically, block-level adaptation contributes +2.1% with 50% fewer parameters (largest gain), spectral-aware regularization and fine-grained routing introduce consistent improvements, together improving from 61.1% to 63.9% (+2.8% overall). The improvements are consistent across vision (+2.6% on DOTA/iNat2018) and NLP (+2.9% on Dolly-15k/BBH, Table 2). Finally, we analyze performance across task difficulty levels and find **3) mtLoRA consistently outperforms SOTA across all difficulty levels.** On BBH, we categorize 27 tasks by average accuracy: Easy (>50%), Medium (30-50%), and Hard (<30%). mtLoRA consistently outperforms SOTA across all levels: +1.6% on Easy, +3.5% on Medium, and +0.4% on Hard tasks (Table 6), demonstrating broad applicability across diverse task difficulties. Remarkably, these gains come with **4) improved parameter and training efficiency**: due to the block-level adaptation design, our $\mathtt{mtLoRA}$ achieves +2.8% performance using only 47% parameters and 24% less training time (Section 4.5).

Our contributions are three-fold. 1) We provide the first systematic analysis of why existing multi-task LoRA methods fail at **scale**. We identify that inter-task alignment (shared knowledge) concentrates in high-SV components; uniform regularization disrupts this, explaining the fundamental regularization-routing trade-off that prior work overlooked. 2) We make three key technical contributions in mechanistic understanding. Spectral-aware regularization selectively orthogonalizes low-SV noise while preserving high-SV shared knowledge, fine-grained routing assigns dimension-specific weights instead of scalar weights, and block-level adaptation mitigates gradient conflict amplification while using 50% fewer parameters. 3) We demonstrate consistent improvements at large-scale (15-25 tasks) across both vision (DOTA, iNat2018) and language (Dolly-15k, BBH) benchmarks, achieving up to 2.8% absolute performance improvement over state-of-the-art while using 47% fewer parameters and 24% less training time, making scalable multi-task LoRA practical for real-world deployments.

## 2    RELATED WORKS

**Multi-Task LoRA Adaptation.**    Multi-task low-rank adaptation (LoRA) aims to compose multiple low-rank updates (Hu et al., 2021) to handle various tasks, simultaneously (Caruana, 1997). The key challenge is the misalignment between low-rank updates (*i.e.*, LoRA experts). Existing solutions can be categorised into regularization and routing methods, respectively tackling the parameter and representation misalignment.

**1) Regularization methods** address the *parameter misalignment*. Existing methods impose regularization to enforce orthogonality across LoRA parameters (Ilharco et al., 2022; Yadav et al., 2023; Yu et al., 2024a). For instance, Task Arithmetic (Ilharco et al., 2022) linearly combined task vectors; TIES-Merging (Yadav et al., 2023) resolved sign conflicts through majority voting; and DARE (Yu et al., 2024a) applied stochastic masking to enforce sparsity. However, these methods are input-independent and ignore input dynamics.

**2) Dynamic routing methods** address *representation misalignment*. These methods typically learn networks to route LoRA experts with learned weights. For instance, hard gating networks (*i.e.*, selecting the top-K LoRAs), and soft routing networks (*i.e.*, combining all LoRA modules with weights). MoLE (Wu et al., 2024a) extended this to LoRA adaptation, introducing Top-K routing and balancing losses to prevent imbalanced expert selection. HydraLoRA (Tian et al., 2024a) combined routing with an asymmetric LoRA structure (*i.e.*, a single shared $A$, with multiple task-specific $B_k$). LoRAMoE forced some LoRA experts to maintain the foundation model's knowledge to protect against catastrophic forgetting. Recent work Hu et al. (2025) resolves task conflicts in representation space, but on hard-parameter shared backbones.

However, these approaches typically treat regularization and routing as independent solutions. As our preliminary study reveals, there is a fundamental trade-off between regularization and routing, which hinders task scalability. Our work is the first to identify and resolve this trade-off, enabling *efficient* and *scalable* multi-task low-rank adaptation.

**Multi-Task LoRA Placement Strategies.**    In multi-task low-rank adaptation, prior works explored where to plug the LoRA modules into transformers. Ada-Merging (Yang et al., 2024) and MoLE (Wu et al., 2024a) found that uniform treatment across layers is suboptimal, so they assigned different weights for LoRAs in different layers. MTLoRA (Agiza et al., 2024) placed task-irrelevant modules at shallow layers and task-specific modules at deep layers in the network. MixLoRA (Wu et al., 2024b) only inserted LoRA into FFN blocks, avoiding attention layers completely. HydraLoRA (Tian et al., 2024a) applied LoRA only to Q and V projection layers (*i.e.*, $W_q$ and $W_v$). However, all these methods apply LoRA to individual, component-level weight matrices ($W_q$, $W_v$, or linear layers within the FFN block). In contrast, we apply LoRA at the block level, as a parallel adapter around attention and FFN blocks. This approach decouples the LoRA update path from the main block's internal computations, hence mitigating the amplification of gradient conflicts.

## 3    METHOD

In this section, we detail our three designs in tackling the *scalability* challenge of multi-task low-rank adaptation. Firstly, we formulate this task, *scalable* multi-task low-rank adaptation, in Sec. 3.1. Then, we detail the three novel designs in mtLoRA: spectral-aware regularization in Sec. 3.2, fine-grained routing in Sec. 3.3, and block-level adaptation in Sec. 3.4. We illustrate the overview of our architectural innovations in Fig. 2.

### 3.1    TASK FORMULATION

We formulate the challenge of *scalable* multi-task low-rank adaptation. Specifically, consider a frozen pretrained model with parameters $W^{(0)}$ and a set of $N$ tasks, we introduce $N$ low-rank updates $\{\Delta_i\}_{i=1}^N$ to the model, where each $\Delta_i(x) = B_i A_i x$ is parameterized by down-projection $A_i \in \mathbb{R}^{r \times d}$ and up-projection $B_i \in \mathbb{R}^{d \times r}$ with rank $r \ll d$ (hidden dimension)[1]. During inference, the multi-task

---

[1]HydraLoRA (Tian et al., 2024a) shows that each low-rank update, especially $B$ matrices, implicitly encodes task-specific knowledge.

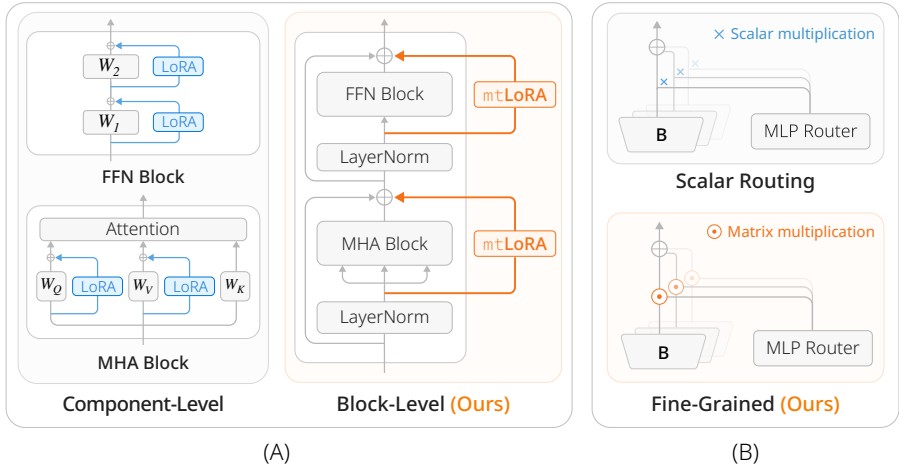

Figure 2: **The architectural innovations of mtLoRA. (A) Block-Level Adaptation.** The LoRA update is computed in a parallel path that bypasses the block's internal non-linearities, mitigating gradient conflict amplification. This path takes the same LayerNorm output as the main block. **(B) Fine-Grained Routing.** Within the parallel path, a router (lightweight MLP) generates dimension-specific weight vectors to compose task experts, allowing different feature subspaces to use different LoRA combinations.

low-rank adaptation combines these low-rank updates:

$$f(x) = f_{W^{(0)}}(x) + \sum_{i=1}^{N} \pi_i(x) \cdot \Delta_i(x). \tag{1}$$

where $\pi_i(x) \in \mathbb{R}$ are scalar routing weights. In this work, we focus on the *scalable* challenge, *i.e.*, $N$ scale-up to a large number, that leads to severe conflicts between low-rank updates. Our three designs address this challenge.

## 3.2 Spectral-Aware Regularization

**Motivation from HydraLoRA's Structure.** Our method builds upon HydraLoRA's asymmetric structure (Tian et al., 2024a), where a shared down-projection $A$ captures task-agnostic representations while multiple up-projections $B_i$ encode task-specific information. Since $A$ is shared, the task conflicts arise from the $B_i$ matrices. Importantly, since $\Delta_i^T \Delta_j = A^T B_i^T B_j A$ with shared $A$, orthogonality between $B_i$ and $B_j$ directly ensures orthogonality between full LoRA updates $\Delta_i$ and $\Delta_j$. As discussed in the Introduction, shared knowledge concentrates in high-SV components, while low-SV shows minimal inter-task alignment. Standard orthogonal regularization forces orthogonality on all components equally, forcing tasks to update in different directions and disrupting knowledge sharing. To address this, we propose spectral-aware regularization: only orthogonalize low-SV components, preserve high-SV shared knowledge. Concretely, for each $B_i \in \mathbb{R}^{d \times r}$, we apply SVD to obtain $B_i = U_i \Sigma_i V_i^T$ with singular values $\{\sigma_k\}$, and construct re-weighted matrices $B_i'$ that emphasize low-SV components via a weighting function $w(\sigma) = \exp(-\sigma/\bar{\sigma})$, where $\bar{\sigma}$ is the mean singular value. Note that this weighting is continuous and adapts to each $B_i$ matrix—the percentile bands (*e.g.*, top-20%) mentioned in our analysis (Fig. 1(B)) are for visualization only, not fixed thresholds in the implementation. The spectral-aware loss is:

$$\mathcal{L}_{\text{spectral}} = \lambda \sum_{i<j} \|(B_i')^T B_j'\|_F^2. \tag{2}$$

This loss encourages orthogonality primarily among the low-SV components, while preserving the high-SV shared knowledge essential for multi-task learning.

## 3.3 Fine-Grained Routing

Unlike conventional dynamic routing that assigns one scalar weight per LoRA ($\pi_i \in \mathbb{R}$), we assign dimension-specific weights. We partition the feature dimension $d$ into $g$ groups, where $g$ denotes

the number of groups (not group size). Our router network outputs a weight vector $\Pi_i \in \mathbb{R}^g$ for each LoRA module $i$ using softmax normalization (soft routing), as shown in Fig. 2(B). For example, $g$=1 corresponds to module-wise routing (one scalar weight per LoRA), while $g$=768 provides full dimension-level routing (one weight per dimension).[2] The low-rank update in Eq. 1 becomes:

$$\sum_{i=1}^{N} \Pi_i(x) \odot \Delta_i(x), \tag{3}$$

where $\odot$ denotes grouped element-wise multiplication. This allows different feature subspaces to use different LoRA combinations. To prevent routing collapse, where the router favors only a few experts, we add a load-balancing auxiliary loss $\mathcal{L}_{\text{balance}}$ to encourage a uniform distribution of routing weights across all experts (Wu et al., 2024a). The total loss is formulated as $\mathcal{L} = \mathcal{L}_{\text{task}} + \lambda_1 \mathcal{L}_{\text{spectral}} + \lambda_2 \mathcal{L}_{\text{balance}}$.

### 3.4 BLOCK-LEVEL ADAPTATION

Instead of adapting individual weight matrices ($W_q, W_v$) inside blocks, we adapt LoRA at the block-level by adding it as a parallel path after attention or FFN blocks. As illustrated in Fig. 2(A), for a frozen block $W^{(F)}$ (*e.g.*, Multi-Head Attention or FFN), the adapted output is:

$$x' = x + W^{(F)}\left(\text{LN}(x)\right) + \Delta\left(\text{LN}(x)\right), \tag{4}$$

where $\Delta = \sum_{i=1}^{N} \Pi_i \odot \Delta_i$ is the combined low-rank update, and LN is the LayerNorm[3]. In this way, the LoRA update path is decoupled from the internal non-linearities (*e.g.*, Softmax) of the main blocks, hence mitigating the amplification of gradient conflict.

***Why does block-level adaptation work?*** Compared with conventional LoRA, the block-level adaptation avoids gradient conflicts in attention. Specifically, traditional LoRA is attached to linear layers (weight matrices $W_q, W_v, W_1, W_2$). The gradients will flow through the Softmax in attention, as shown in Figure 2(A). This creates cross-token dependencies: changing attention to one token affects all other token positions. This effect amplifies task conflicts. For example, consider the input "The bank is steep". For finance tasks, the model needs high attention on "bank"→"money". For geography tasks, the model needs "bank"→"river". These conflicting attention patterns interfere through Softmax. In traditional LoRA, updating $B$ to increase "bank"→"money" attention automatically decreases "bank"→"river" attention due to Softmax normalization, as they compete for the same probability mass. Our block-level adaptation avoids this competition. The two adapters can add the "money" and "river" feature **independently** to "bank" representation.

## 4 EXPERIMENTS

We first provide an experimental setup in Sec 4.1, and then reveal the core challenge (*i.e.*, multi-task collapse) in a preliminary study in Sec 4.2. Given this understanding, we ablate the working mechanisms of our three key designs in Sec 4.3, and perform an SOTA comparison in Sec 4.4. Additionally, we discuss the applicability and limitations of our designs in Sec 4.5.

### 4.1 EXPERIMENTAL SETUP

**Benchmarks.** We evaluate `mtLoRA` on four benchmarks: DOTA (Xia et al., 2018) (15 tasks), iNat2018 (Van Horn et al., 2018) (25 tasks), Dolly-15k (Conover et al., 2023) (16 tasks) with evaluation on MMLU, and Flanv2 (Longpre et al., 2023) subset with evaluation on Big Bench Hard (BBH) (Suzgun et al., 2022). We compare with state-of-the-art methods including HydraLoRA (Tian et al., 2024a), MMoELoRA (Wu et al., 2024a), and LoRAHub (Huang et al., 2023), with LoRA rank $r = 16$. Detailed experimental settings and compared methods' implementations are provided in Appendix B.2.

---

[2] The router outputs $g$ weights per LoRA. For intermediate $g$ (*e.g.*, $g$=32), each weight is repeated $d/g$ times to cover its dimension group.

[3] We apply LoRA after LayerNorm, consistent with the Pre-LayerNorm (Pre-LN) architecture of ViT (Dosovitskiy et al., 2020) and LLama2-7B (Touvron et al., 2023).

**Implementation of Our Method.** Our method is based on HydraLoRA (Tian et al., 2024a) architecture, where a single matrix $A$ is shared across all tasks, and tasks are learned through diverse $B_i$ matrices. Specifically, we apply SVD to the task-specific $B$ matrices once per epoch to compute $\mathcal{L}_{\text{spectral}}$ for efficiency. This structure allows us to directly regularize spectrums of $B_i$ (instead of entire LoRA) to control task-specific conflicts. We apply SVD to the task-specific $B$ matrices once per epoch to compute the spectral-aware loss. The loss is $\mathcal{L}_{\text{spectral}} = \lambda \sum_{i<j} \|(B_i')^T B_j'\|_F^2$, where $B_i' = U_i \Sigma_i' V_i^T$ is a temporary matrix constructed by re-weighting the singular values of $B_i = U_i \Sigma_i V_i^T$ with $\Sigma_{kk}' = \sqrt{w(\sigma_k)} \cdot \sigma_k$ where $w(\sigma) = \exp(-\sigma/\bar{\sigma})$ to penalize low-SV components. The router is a 2-layer MLP with output dimension $N \times g$, where $N$ is the task number and $g$ is the number of groups. It takes the mean-pooled hidden states as input and applies softmax normalization to produce routing weights. For $g > 1$, each weight is broadcast by repeating $d/g$ times before element-wise multiplication with LoRA outputs. The total loss includes a load-balancing term $\mathcal{L}_{\text{balance}}$ to prevent routing collapse. All LoRA modules are applied at the block level as parallel adapters, consistent with a Pre-LN architecture (such as AdaptFormer (Chen et al., 2022)).

## 4.2 CHALLENGE OF MULTI-TASK COLLAPSE

We quantitatively analyze the multi-task collapse in Appendix C.1 Table S3 (due to page limit). Results show that performance degrades catastrophically across all datasets: 88.2%→2.0% from 5 to 15 tasks on DOTA, 87.0%→0.5% from 1 to 100 tasks on iNat2018, and 46.1%→16.0% from 4 to 16 tasks on Dolly-15k. Meanwhile, the conflict score (see Appendix B.2 for calculating details) reaches 97.9%, 99.7%, and 64.7%, respectively.

## 4.3 ABLATION STUDIES OF OUR METHOD

We conduct ablation studies to reveal how our three designs work. We answer three key questions:

**Q1: How does spectral-aware regularization resolve the routing-regularization trade-off?** Figure 1(A) shows that when training LoRA modules with dynamic routing on a large-scale multi-task scenario (Flan-v2 training evaluated on BBH, with 16 experts), strong orthogonal regularization ($\lambda = 1.0$) degrades accuracy despite reducing conflicts. *Why?* Table 1 shows orthogonal regularization achieves only 20.5% on DOTA without routing but jumps to 89.8% with dynamic routing. Without routing, all tasks contribute equally (uniform $1/N$ weighting), causing orthogonalized parameters to cancel out. Dynamic routing selectively activates task-relevant LoRAs, making orthogonalization beneficial.

However, even with routing, stronger regularization ($\lambda = 1.0$) degrades performance by 1.8% compared to moderate regularization ($\lambda = 0.25$). Figure 1(A) shows routing uncertainty (entropy) increases from 2.62 to 2.67, indicating the router becomes less decisive. Furthermore, we observe that

Table 1: **Orthogonal regularization requires dynamic routing.** Without routing (uniform $1/N$ weighting), orthogonal regularization achieves only 20.5% on DOTA; with dynamic routing, it improves to 89.8%. Sparsity regularization harms performance in both settings. All results in accuracy (%)↑.

| Method | DOTA | iNat2018 | Avg. |
|---|---|---|---|
| Uniform Routing[†] | | | |
|     HydraLoRA | 18.0 | 8.5 | 13.3 |
|     + Sparsity Reg. | 16.5 | 7.2 | 11.9 |
|     + Orthogonal Reg. | 20.5 | 10.1 | 15.3 |
| Dynamic Routing | | | |
|     HydraLoRA | 89.0 | 78.3 | 83.8 |
|     + Sparsity Reg. | 87.9 | 77.2 | 82.6 |
|     + Orthogonal Reg. | **89.8** | **79.8** | **84.8** |

[†]Uniform weighting: $1/N$ per LoRA.

reaching the optimal point at $\lambda = 0.25$ requires approximately $1.4\times$ more training iterations compared to no regularization, suggesting that naive orthogonality makes the optimization landscape more challenging. The core of multi-task learning is to share underlying knowledge across tasks. Uniform orthogonalization forces tasks to update in different directions, directly disrupting this knowledge sharing. This motivates spectral-aware regularization: preserve high-SV shared knowledge, only orthogonalize low-SV components. Figure 1(B) shows top-20% singular values have 89% inter-task alignment and encode 54% of total singular values (vs. 3% alignment and 22% for bottom-50%). Our spectral-aware approach only orthogonalizes low-SV components, preserving the high-SV shared knowledge needed for effective routing. To empirically validate this selective effect, we visualize

Table 2: **Contribution of each key design.** `mtLoRA` improves +2.8% over baseline with 47% fewer parameters (highlighted in blue ). Block-level adaptation contributes the most (+2.1%). Improvements are consistent across vision (+2.6% by average) and NLP (+2.9% by average) benchmarks. All results in average accuracy (%)↑, reported with std across 3 random seeds. Params shown as trainable parameters and % of LLaMA-2-7B. Wall-clock time breakdown in Appendix C.6.

| Method | Block-Level Adaptation | Spectral-Aware Regularization | Fine-Grained Routing | Params (%) | Time | DOTA | iNat2018 | Dolly-15k | BBH | Avg. |
|---|---|---|---|---|---|---|---|---|---|---|
| HydraLoRA | | | | 75.5M (1.11%) | 1.00x | 89.0±0.4 | 78.3±1.7 | 41.6±1.0 | 35.5±1.7 | 61.1 |
| mtLoRA (Ours) | ✓ | | | 37.7M (0.56%) | 0.67x | 91.2±0.2 | 79.9±1.0 | 43.7±0.4 | 37.9±0.4 | 63.2 |
| | ✓ | ✓ | | 37.7M (0.56%) | 0.70x | **91.7**±0.4 | 81.3±1.1 | 43.6±0.4 | 38.4±0.3 | 63.8 |
| | ✓ | | ✓ | 39.8M (0.59%) | 0.69x | 89.9±0.5 | 80.2±0.7 | 44.1±0.3 | 38.2±0.2 | 63.1 |
| | ✓ | ✓ | ✓ | 39.8M (0.59%) | 0.76x | 91.0±0.8 | **81.5**±0.6 | **44.5**±0.3 | **38.5**±0.3 | **63.9** |

Table 4: **Ablation of block-level adaptation.** Block-level achieves better performance with **50% fewer parameters** and **33% less wall-clock time**. Specifically, Attn+FFN achieves 63.1% average accuracy (+2.0% over component-level) with same parameter count (75.5M, 1.1% of foundation model). Notably, FFN alone uses **only 50% parameters** (37.7M vs 75.5M) yet achieves 63.0% average (highlighted in blue ). All results in accuracy (%)↑. Params shown as trainable parameters and % of LLaMA-2-7B. Wall-clock time breakdown in Appendix C.6.

| Adaptation Level | | Params (%) | Time | DOTA | iNat2018 | Dolly-15k | BBH | Avg. |
|---|---|---|---|---|---|---|---|---|
| Component-Level | | | | | | | | |
| $W_q, W_v$ | | 75.5M (1.11%) | 1.00x | 89.0 | 78.3 | 41.6 | 35.5 | 61.1 |
| Block-Level | | | | | | | | |
| Attn | FFN | | | | | | | |
| ✓ | | 37.7M (0.56%) | 0.67x | 89.4 | 79.3 | 43.3 | 37.2 | 62.3 |
| | ✓ | **37.7M (0.56%)** | 0.67x | 91.0 | 79.4 | 43.7 | **37.9** | 63.0 |
| ✓ | ✓ | 75.5M (1.11%) | 0.85x | **91.2** | **79.9** | **43.9** | 37.6 | **63.1** |

the SV spectrum before/after regularization in Figure S2 in Appendix. Results confirm that low-SV components are suppressed 3× more (−6.0%) than high-SV components (−2.0%).

**Q2: How does fine-grained routing exploit dimension-specific heterogeneity?** Standard multi-task routing assigns one scalar weight per LoRA module ($g$=1 group), forcing all dimensions to use the same task mixture. As discussed in Sec. 3.3, different feature dimensions encode different task attributes (*e.g.*, creativity vs. accuracy). Coarse-grained routing ($g$=1) applies uniform weights to all dimensions, ignoring this heterogeneity. In contrast, fine-grained routing (larger $g$) assigns dimension-specific weights to capture diverse attribute requirements.

Table 3 shows fine-grained routing ($g$=32) achieves 37.7% on BBH, outperforming module-wise routing ($g$=1, 35.5%) by +2.2%. The benefits vary by task type: reasoning tasks (BBH) benefit significantly (+2.2%), while instruction-following tasks (Dolly-15k) show modest gains, suggesting reasoning tasks exhibit stronger dimension-specific heterogeneity. Notably, $g$=2 already provides **+1.5% improvement with only +0.06% ex-tra router parameters**, offering a strong efficiency-performance trade-off. In implementation, we can select routing granularity based on their resource constraints: $g$=2 for minimal overhead, $g$=32 for maximum performance.

Table 3: **Ablation of routing granularity.** Fine-grained routing balances performance and parameter efficiency. The $g$=2 already achieves +1.5% with only +0.06% extra router parameters (highlighted in blue ), and $g$=32 achieves best performance (+2.2% on BBH). Results in accuracy (%)↑. Router $\Delta$ shown as extra param % of LLaMA-2-7B.

| Strategy | $g$ | Dolly-15k | BBH | Avg. | Router $\Delta$ |
|---|---|---|---|---|---|
| Module-Wise | | | | | |
| Scalar | 1 | 41.6 | 35.5 | 38.5 | — |
| Fine-Grained | | | | | |
| | 2 | 41.6 | 37.0 | 39.3 | +0.06% |
| | 8 | 41.3 | 36.6 | 39.0 | +0.44% |
| Grouped | 16 | 41.7 | 37.0 | 39.3 | +0.93% |
| | 32 | **42.0** | **37.7** | **39.9** | +1.93% |

**Q3: How does block-level adaptation reduce gradient conflicts?** As detailed in Sec. 3.4,

Table 5: **Comparison with SOTA.** mtLoRA achieves 64.0% average accuracy across four benchmarks (91.7% on DOTA, 81.5% on iNat2018, 44.5% on Dolly-15k, 38.5% on BBH), outperforming previous SOTA HydraLoRA by 2.3% on average. We compare with multiple SOTA methods including LoRAHub, MMoELoRA, and HydraLoRA with identical experimental setup (rank $r = 16$, with identical experts numbers). Results obtained with rank $r = 16$. All results in average accuracy (%)↑.

| Method | DOTA | iNat2018 | Dolly-15k | BBH | Avg. |
|---|---|---|---|---|---|
| LoRAHub (Huang et al., 2023) | 88.9±1.7 | 80.2±1.6 | 42.0±0.3 | 34.9±0.4 | 61.5 |
| MMoELoRA (Zadouri et al., 2023) | 89.4±0.2 | 78.0±0.3 | 42.1±0.8 | 35.4±0.9 | 61.2 |
| HydraLoRA (Tian et al., 2024a)[‡] | 89.1±0.4 | 78.5±1.7 | 42.4±0.7 | 36.9±1.0 | 61.7 |
| mtLoRA (Ours) | **91.7**±0.4 | **81.5**±0.6 | **44.5**±0.2 | **38.5**±0.3 | **64.0** |

[‡] *Implemented with optimal hyperparameter search and BLC optimization.*

component-level adaptation ($W_q, W_v$) suffers from gradient conflict amplification: gradients propagate through attention's Softmax, creating cross-token dependencies. Instead, our block-level adaptation learns a residual $\Delta(h)$ that bypasses these internal non-linearities, decoupling the adaptation process from conflict-prone attention mechanics. Table 4 shows block-level adaptation (Attn+FFN) achieves 63.1% average accuracy (+2.0% over component-level), revealing a consistent trend: component (61.1%) → attention (62.3%) → FFN (63.0%) → Attn+FFN (63.1%). Notably, FFN alone uses only 50% parameters (37.7M vs 75.5M) yet achieves 63.0% average, demonstrating strong parameter efficiency.

*Gradient misalignment.* To verify that block-level adaptation actually reduces gradient conflicts, we measure gradient alignment between task pairs. We compare component-level adaptation ($W_q, W_v$) and block-level adaptation on LLaMA2-7B using Dolly-15k (16 tasks). For 5,000 iterations, we sample two tasks (A, B) and one data point from each. With the base model frozen, we compute loss and extract LoRA gradient vectors $\nabla W_A$ and $\nabla W_B$ via backward pass. We compute cosine similarity $\cos(\nabla W_A, \nabla W_B)$, where negative values indicate conflict. As shown in Figure 1(C), component-level adapters show a mean cosine similarity of -0.054±0.201, indicating substantial misalignment. In contrast, block-level adapters achieve -0.013±0.169. This represents a 76% reduction in the magnitude of average gradient conflict, creating an easier optimization landscape. Additionally, we provide per-layer gradient correlation analysis in Appendix Section C.3 (Figure S3), showing up to 36% conflict reduction in later layers.

## 4.4 COMPARISON WITH STATE-OF-THE-ART

In this section, we compare the overall effectiveness with SOTA methods. Specifically, we compare with three SOTA multi-task low-rank adaptation approaches, HydraLoRA, MMoELoRA, LoRAHub, and analyze the contribution of each component through ablation. To ensure fair comparison, all compared methods use the identical experimental setup: same training hyperparameter configs (*e.g.*, rank $r = 16$), and same LoRA numbers per benchmark. The comparison results are shown in Tables 5 and 2.

Based on comprehensive comparison against multiple SOTA methods (LoRAHub, MMoELoRA, and HydraLoRA) with identical experimental setup, we have three key findings: **1) Our method outperforms SOTA at scale.** mtLoRA achieves 64.0% average accuracy across four benchmarks (91.7% on DOTA, 81.5% on iNat2018, 44.5% on Dolly-15k, 38.5% on BBH), outperforming previous SOTA HydraLoRA by 2.3% on average (Table 5). This demonstrates that our method effectively mitigates the multi-task collapse problem, enabling scalable adaptation to a larger number of tasks. **2) Existing methods fail to scale beyond 15 tasks.** LoRAHub and MMoELoRA achieve average accuracy around 61% (61.5 and 61.2 respectively), despite being designed for smaller-scale scenarios with 5-8 tasks. Single LoRA achieves 94.5% on DOTA, establishing the upper bound without multi-task learning. This highlights the challenge of multi-task low-rank adaptation: existing methods cannot bridge the gap between single-task and multi-task performance. **3) Each component contributes substantially.** Results show that all three designs are instrumental: Block-level adaptation contributes +2.1% with 50% fewer parameters (Table 4), spectral-aware regularization and fine-grained routing contribute consistent improvements. The improvements are consistent across vision (+2.6%) and NLP (+2.9%), together improving from 61.1% to 63.9% (+2.8% overall, Table 2).

Table 6: **Performance breakdown by task difficulty on BBH.** Tasks' difficulty are categorized by average accuracy across methods: Easy (>50%), Medium (30-50%), Hard (<30%) (category details in Appendix C.5). mtLoRA achieves best across all difficulty levels, especially on Medium tasks (+3.5% over HydraLoRA, highlighted ). All results in accuracy (%)↑. Per-task results in Table S6.

| Difficulty | LoRA | MMoELoRA | HydraLoRA | mtLoRA |
|---|---|---|---|---|
| Easy (7 tasks) | 64.32 | 63.74 | 67.96 | **69.52** |
| Medium (8 tasks) | 37.82 | 40.82 | 37.55 | **41.01** |
| Hard (12 tasks) | 14.63 | 15.20 | 18.39 | **18.78** |
| Overall (27 tasks) | 34.38 | 35.37 | 36.92 | **38.52** |

## 4.5 DISCUSSION

**A) Consistent Improvement Across Task Difficulty.** To understand where mtLoRA excels, we categorize BBH's 27 tasks by difficulty based on average accuracy: Easy (>50%), Medium (30-50%), and Hard (<30%). As shown in Table 6, mtLoRA consistently outperforms SOTA across all difficulty levels: +1.6% on Easy, +3.5% on Medium, and +0.4% on Hard tasks, demonstrating broad applicability rather than being limited to specific task regimes. Per-task breakdown results on BBH are provided in Table S6 in Appendix.

**B) Domain Differences.** Block-level adaptation universally improves both vision (+2.1%) and NLP (+2.3%) domains, while fine-grained routing shows dataset-dependent effects (-1.3% on DOTA). See Appendix C.4 for detailed analysis. This suggests that gradient conflict mitigation is a universal challenge, while dimension-specific routing benefits depend on feature heterogeneity.

**C) Computational Efficiency.** As shown in Table 2, our mtLoRA achieves +2.8% performance improvement while simultaneously being more efficient: 47% fewer parameters and 24% faster training. Block-level adaptation is the key contributor, reducing training time by 33% (94.6 min $\rightarrow$ 63.0 min) with 50% fewer parameters (75.5M $\rightarrow$ 37.7M). Notably, FLOPs reduction is modest (0.85x-0.99x), indicating that speedup primarily comes from improved GPU utilization rather than reduced computation. See Appendix C.6 for detailed breakdown.

**D) Limitations and Future Work.** Our design choices entail certain trade-offs. 1) We use a **shared matrix 'A'** structure for efficiency, reducing SVD cost from $O(d^3)$ to $O(dr^2)$. While efficient, our core designs (spectral-aware regularization, fine-grained routing, block-level adaptation) are generalizable to standard LoRA arch ($\Delta_i = B_i A_i$) with minimal modification. 2) While our **block-level adaptation** targets Transformers, the underlying principle, *i.e.*, bypassing conflict-amplifying non-linearities such as Softmax, is architecture-agnostic and extends to others like CNNs. We explore the applicability to larger models (*e.g.*, LLaMA2-13B) in the Supplementary. Finally, our **working hypothesis**, *i.e.*, shared knowledge concentrates in high-SV while low-SV shows minimal inter-task alignment, is empirically supported by our observations (Figure 1(B)) and prior works (Yu et al., 2024b; 2025). Its generalizability across other modalities warrants future investigation.

**LLM Usage Claim.** We employed Large Language Models (LLMs) for partial text polishing, listing related papers, and optimizing experimental deployment scripts. However, we confirm the core methodological innovations, critical code implementation, interpretation of results, and final manuscript verification remain authors' sole work.

## 5 CONCLUSION

We present mtLoRA, enabling stable, scalable multi-task adaptation by addressing the limitations of prior approaches. We identify that existing methods fail due to the regularization-routing trade-off, rooted in spectral heterogeneity and gradient conflict amplification. Motivated by insights that inter-task alignment (shared knowledge) concentrates in high-SV components, and that component-level adaptation amplifies conflicts, we propose three designs: spectral-aware regularization, fine-grained routing, and block-level adaptation. Our approach achieves up to 2.8% improvement over SOTA while using 47% fewer parameters and 24% less training time across vision and NLP benchmarks, offering a parameter-efficient, compute-efficient, and robust path for scalable multi-task adaptation.

ACKNOWLEDGEMENTS

The authors gratefully acknowledge the support from the DSO research grant awarded by DSO National Laboratories, Singapore. This project is also partially supported by the Ministry of Education, Singapore, under its Tier-1 Academic Research Fund (No. 24-SIS-SMU-040).

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

SUPPLEMENTARY MATERIAL

This supplementary material provides additional theoretical foundations, method implementation details, and extended experimental results that complement our main manuscript (references to the main manuscript are shown in red).

**A. Theoretical Foundations**

- Section A.1 provides mathematical justification for Gradient Conflict in attention mechanisms (Section 3.3).

**B. Implementation Details**

- Section B.1 illustrates the detailed architecture of mtLoRA (Figure 2).
- Section B.2 provides detailed experimental setups, including dataset construction, metric definitions, and implementation details (Section 4.1).

**C. Supplementary Experimental Results**

- Section C.1 provides additional analysis on the multi-task collapse challenge (Section 4.2).
- Section C.2 visualizes the effect of spectral-aware regularization (Section 4.3 Q1).
- Section C.3 provides per-layer gradient correlation analysis for block-level adaptation (Section 4.3 Q3).
- Section C.4 analyzes domain differences between vision and NLP tasks (Section 4.5).
- Section C.5 presents detailed per-task results and difficulty analysis on 27 BBH reasoning tasks (Table 4).
- Section C.6 provides computational efficiency analysis with wall-clock time breakdown (Section 4.5C, Tables 3-4).

# A  THEORETICAL FOUNDATIONS

## A.1  JUSTIFICATION OF GRADIENT CONFLICT

**Mathematical Analysis of Gradient Conflict (Section 3.4).**  Define gradient conflict as the expected negative cosine similarity between task gradients:

$$\mathcal{C} = \mathbb{E}_{t_1, t_2} \left[ -\cos \left( \nabla_B^{t_1}, \nabla_B^{t_2} \right) \right] \tag{5}$$

For traditional LoRA, the gradient includes the Softmax Jacobian $J_{\text{SM}}$:

$$\nabla_{B_q}^t = J_{\text{SM}} \times \nabla_{\text{Attn}}^t \times (A_q h) \tag{6}$$

where $J_{\text{SM}}[i,j] = S_i(\delta_{ij} - S_j)$ creates off-diagonal coupling. This coupling amplifies conflicts: even if $\nabla_{\text{Attn}}^{t_1}$ and $\nabla_{\text{Attn}}^{t_2}$ have localized differences, $J_{\text{SM}}$ spreads them across all positions. Our residual adapter eliminates this amplification by bypassing $J_{\text{SM}}$ entirely.

# B  IMPLEMENTATION DETAILS

## B.1  ARCHITECTURE DETAILS

Figure S1 provides the detailed architecture of mtLoRA (Figure 2). The architecture shows how our three key designs are integrated into a standard Transformer block. Specifically, mtLoRA modules are attached in parallel paths after each LayerNorm, bypassing the internal non-linearities of the frozen blocks to mitigate gradient conflict amplification.

## B.2  EXPERIMENTAL SETUP (SECTION 4.1)

**Benchmarking Datasets.**  We evaluate multi-task low-rank adaptation on four benchmarks, *i.e.*, DOTA (Xia et al., 2018) (15 cross-domain tasks), iNat2018 (Van Horn et al., 2018) (25-100 fine-grained classification tasks), Dolly-15k (Conover et al., 2023) (16 instruction-following tasks), and

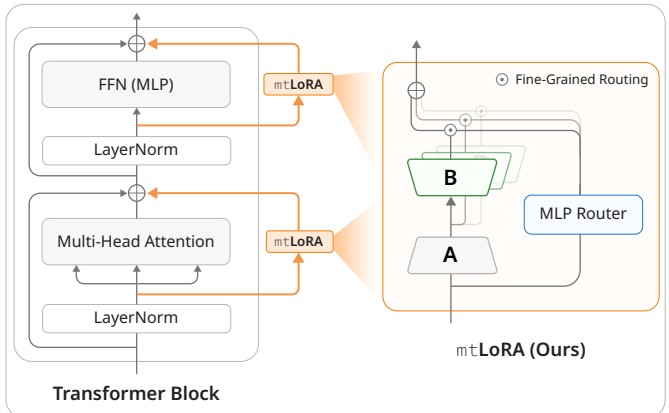

Figure S1: **Overall Architecture of mtLoRA. Left:** A Transformer block consists of Multi-Head Attention and FFN (MLP) components, each preceded by LayerNorm. mtLoRA modules are attached in parallel paths after each LayerNorm. **Right:** The internal structure of mtLoRA shows the fine-grained routing mechanism, where a Router MLP generates dimension-specific weights to dynamically compose task-specific experts.

Big Bench Hard (BBH) (Suzgun et al., 2022). For iNat2018, we partition fine-grained categories to construct 25 tasks with high visual similarity. For Dolly-15k, which contains only 5 tasks, we apply K-Means clustering on instruction embeddings to partition data into 16 semantically distinct clusters, treating each cluster as a task (Tian et al., 2024a). This evaluation protocol follows HydraLoRA's setup (Tian et al., 2024a). Besides, for naturally-defined tasks, our method is further validated on three benchmarks (DOTA, iNat2018, BBH). For BBH, we evaluate the model's generalization capability on complex reasoning tasks. Specifically, we construct a multi-task training set from Flan-v2 (Longpre et al., 2023) by sampling 30,000 examples evenly across 10 diverse task clusters (*e.g.*, commonsense reasoning, translation, QA). After multi-task training, we evaluate the model on all 27 BBH reasoning tasks using 3-shot in-context learning. This setup tests whether mtLoRA can effectively transfer capabilities from instruction tuning to challenging reasoning tasks.

**Evaluation Metrics.** To quantitatively analyze the challenges in multi-task low-rank adaptation, we define three key metrics. 1) **Gradient Conflict Score**: We measure the overall conflict between task pairs by computing the average cosine similarity of their respective LoRA gradient vectors, $\cos(\nabla W_i, \nabla W_j)$, where a more negative value indicates stronger conflict. Note that while this metric specifically detects destructive interference (opposing gradients), our spectral regularization (Eq. 5) penalizes any non-orthogonality (absolute cosine) to minimize both conflict and redundancy. For the spectral analysis in Figure 1(B), to precisely quantify inter-task alignment within a spectral band $B$, we (performed on converged models) use a singular-value-weighted score:

$$\mathcal{A}(B) = \frac{1}{|B|N(N-1)} \sum_{k \in B} \sum_{i \neq j} \sigma_{i,k} \sigma_{j,k} |\cos(\mathbf{u}_{i,k}, \mathbf{u}_{j,k})| \tag{7}$$

where $k \in B$ indexes singular value positions, $i, j$ index LoRA modules, $\sigma_{i,k}$ is the $k$-th singular value of module $i$'s $B_i$ matrix, and $\mathbf{u}_{i,k}$ is the corresponding left singular vector. High alignment in high-SV reflects shared knowledge across tasks. 2) **Spectral Band Contribution**: By definition of SVD, larger singular values correspond to principal directions of task-specific parameter updates, so a higher fraction indicates more task-relevant information. To quantify task-specific information within a spectral band (Figure 1(B)), we compute the fraction of total singular values: $\sum_{k \in B} \sigma_k / \sum_{\text{all}} \sigma_k \times 100\%$, where $B$ denotes the spectral band and $\sigma_k$ are the singular values of the $B_i$ matrices. 3) **Routing Uncertainty**: To measure the router's decision-making confidence for each input, we calculate the average per-sample routing entropy: $\mathbb{E}_x \left[ -\sum_{i=1}^{N} \pi_i(x) \log \pi_i(x) \right]$, where $\pi(x)$ is the router's output distribution for a given sample $x$. We report routing uncertainty (entropy); lower values indicate more confident, decisive routing.

**Implementation of Our Method.** Our method is based on HydraLoRA (Tian et al., 2024a), where a single matrix $A$ is shared across all tasks, and tasks are learned through diverse $B_i$ matrices. Since

Table S1: **Computational efficiency breakdown for main ablation** (Table 2). Block-level adaptation reduces training time by 33% while using 50% fewer parameters. Full `mtLoRA` is 24% faster than HydraLoRA with 47% fewer parameters.

| Method | Block | Spec. | FGR | Params | % | Time (min) | Rel. Time | FLOPs | Rel. FLOPs |
|---|---|---|---|---|---|---|---|---|---|
| HydraLoRA | | | | 75.5M | 1.11% | 94.6 | 1.00× | 4.73e17 | 1.00× |
| `mtLoRA` | ✓ | | | 37.7M | 0.56% | 63.0 | 0.67× | 4.70e17 | 0.99× |
| | ✓ | ✓ | | 37.7M | 0.56% | 66.1 | 0.70× | 4.01e17 | 0.85× |
| | ✓ | | ✓ | 39.8M | 0.59% | 65.5 | 0.69× | 4.70e17 | 0.99× |
| | ✓ | ✓ | ✓ | 39.8M | 0.59% | 72.1 | 0.76× | 4.01e17 | 0.85× |

Table S2: **Computational efficiency breakdown for block-level ablation** (Table 4). FFN-only achieves same efficiency as Attn-only but better performance (63.0% vs 62.3%). Attn+FFN is 15% faster than component-level despite same parameter count.

| Configuration | Params | % | Time (min) | Rel. Time | FLOPs | Rel. FLOPs |
|---|---|---|---|---|---|---|
| Component ($W_q$, $W_v$) | 75.5M | 1.11% | 94.6 | 1.00× | 4.73e17 | 1.00× |
| Block Attn only | 37.7M | 0.56% | 63.4 | 0.67× | 4.70e17 | 0.99× |
| Block FFN only | 37.7M | 0.56% | 63.0 | 0.67× | 4.70e17 | 0.99× |
| Block Attn+FFN | 75.5M | 1.11% | 80.3 | 0.85× | 4.73e17 | 1.00× |

$\Delta_i^T \Delta_j = A^T B_i^T B_j A$, orthogonality between $B_i$ and $B_j$ (*i.e.*, $B_i^T B_j \approx 0$) ensures orthogonality between entire LoRA updates $\Delta_i$ and $\Delta_j$. This structure allows us to directly regularize spectrums of $B_i$ (instead of entire LoRA) to control task-specific conflicts. We apply SVD to the task-specific $B$ matrices per epoch to compute the spectral-aware loss. The loss is $\mathcal{L}_{\text{spectral}} = \lambda \sum_{i<j} \|(B_i')^T B_j'\|_F^2$, where $B_i' = U_i \Sigma_i' V_i^T$ is a temporary matrix constructed by re-weighting the singular values of $B_i = U_i \Sigma_i V_i^T$ with $\Sigma_{kk}' = \sqrt{w(\sigma_k)} \cdot \sigma_k$ where $w(\sigma) = \exp(-\sigma/\bar{\sigma})$ to penalize low-SV components. The router is a 2-layer MLP with output dimension $N \times g$, where $N$ is the task number and $g$ is the number of groups. It takes the mean-pooled hidden states as input and applies softmax normalization to produce routing weights. For $g > 1$, each weight is broadcast by repeating $d/g$ times before element-wise multiplication with LoRA outputs. The total loss includes a load-balancing term $\mathcal{L}_{\text{balance}}$ to prevent routing collapse. All LoRA modules are applied at the block level as parallel adapters, consistent with a Pre-LN architecture (such as AdaptFormer (Chen et al., 2022)).

**Implementation of Compared Methods.** We compare with HydraLoRA (Tian et al., 2024a), MMoELoRA (Wu et al., 2024a), and LoRAHub (Huang et al., 2023). All compared methods use rank $r = 16$ for fair comparison. For experiments involving varying regularization strengths ($\lambda$), we perform hyperparameter search for the optimal learning rate for each $\lambda$ to ensure a fair comparison.

## C  SUPPLEMENTARY EXPERIMENTAL RESULTS

### C.1  ANALYSIS OF MULTI-TASK COLLAPSE

Table S3: **Multi-task collapse increases with task numbers.** Naive averaging degrades from 88.2% (5 tasks) to 2.0% (15 tasks) on DOTA, with conflict score reaching 97.9%. Single LoRA achieves 94.5% on DOTA, 87.0% on iNat2018. All results in accuracy (%)↑.

| | DOTA | | | iNat2018 | | | | Dolly-15k | | |
|---|---|---|---|---|---|---|---|---|---|---|
| | 5 | 10 | 15 | 15 | 25 | 80 | 100 | 4 | 8 | 16 |
| Single LoRA | 94.5 | 94.5 | 94.5 | 87.0 | 87.0 | 87.0 | 87.0 | 45.5 | 45.5 | 45.5 |
| Naive Averaging | 88.2 | 12.0 | 2.0 | 3.5 | 1.0 | 0.5 | 0.3 | 46.1 | 40.5 | 16.0 |
| Conflict Score | 6.7 | 87.3 | 97.9 | 96.0 | 98.9 | 99.4 | 99.7 | −1.5 | 10.9 | 64.7 |

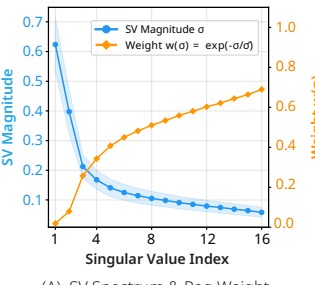 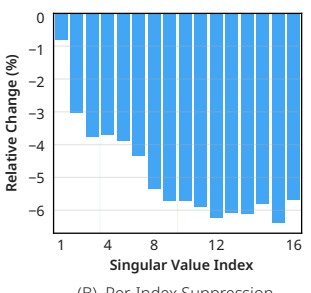 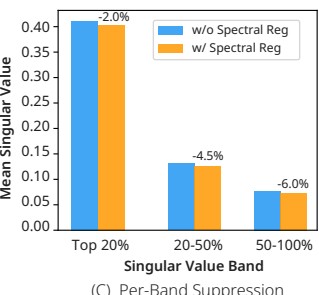

(A) SV Spectrum & Reg Weight          (B) Per-Index Suppression          (C) Per-Band Suppression

Figure S2: **Spectral-aware regularization selectively suppresses low-SV components. (A) Higher weight $\sigma$ on low-SV.** The singular value magnitude $\sigma$ (blue) decays rapidly across indices. Our weighting function $w(\sigma) = \exp(-\sigma/\bar{\sigma})$ (orange) assigns higher regularization weights for low-SV components. **(B) Per-index suppression.** The relative change in SV magnitude after applying spectral regularization ($\lambda = 1.0$). Low-SV components show stronger suppression. **(C) Per-band suppression.** Aggregated comparison across three bands: top-20% ($-2.0\%$), 20–50% ($-4.5\%$), and 50–100% ($-6.0\%$).

Table S3 supplements multi-task collapse analysis (Section 4.2). Results show that naive averaging of experts leads to catastrophic performance degradation as expert numbers increase, accompanied by rising conflict scores.

## C.2   SPECTRAL REGULARIZATION ANALYSIS

Figure S2 visualizes the effect of our spectral-aware regularization (Section 3.2). We analyze the singular value spectrum of $B$ matrices from 512 LoRA modules (across all layers and experts) trained on BBH with LLaMA-2-7B.

**Key Observations.**  1) **Selective suppression**: Low-SV components (50–100%) are suppressed by 6.0%, while high-SV components (top-20%) are preserved with only 2.0% reduction. This $3\times$ difference confirms that our weighting function $w(\sigma) = \exp(-\sigma/\bar{\sigma})$ selectively targets noise-prone subspaces. 2) **Monotonic effect**: The suppression increases monotonically from high-SV to low-SV ($2.0\% \rightarrow 4.5\% \rightarrow 6.0\%$), validating that our design preserves task-discriminative directions while reducing interference in less informative subspaces. 3) **Consistent with theory**: This empirical result aligns with the Intrinsic Low-Rank Hypothesis—high-SV components encode principal task directions (signal), while low-SV components correspond to optimization noise.

## C.3   PER-LAYER GRADIENT CORRELATION ANALYSIS

Figure S3 visualizes the per-layer pairwise gradient similarity between tasks, comparing component-level (HydraLoRA, $W_q/W_v$) and block-level (mtLoRA, FFN) adaptation (Section 4.3 Q3). We compute the average cosine similarity of task-pair gradients on `lora_B` parameters at each of the 32 transformer layers, using 9 task clusters from Flan-v2 with 30 samples per task.

**Key Observations.**  1) **Block-level reduces overall conflict**: Block-level adaptation achieves lower mean gradient similarity (0.540 vs 0.579, $-6.7\%$), indicating reduced inter-task gradient conflict. 2) **Largest improvement in later layers**: The most significant conflict reduction occurs in later layers—Layer 24 ($-36\%$: 0.752→0.392), Layer 31 ($-37\%$: 0.824→0.516), and Layer 29 ($-39\%$: 0.742→0.451). This suggests that block-level adaptation is particularly effective at isolating task-specific updates in deeper representations. 3) **Early layers show increased similarity**: Conversely, early layers (0, 3, 4) exhibit higher gradient similarity under block-level adaptation, possibly because early layers encode more task-agnostic features where block-level routing introduces shared gradient patterns. This layer-wise heterogeneity suggests that **layer-specific adapter granularity**—applying block-level adapters only to later layers while using component-level adapters for early layers—could further optimize the conflict-performance trade-off, warranting future investigation.

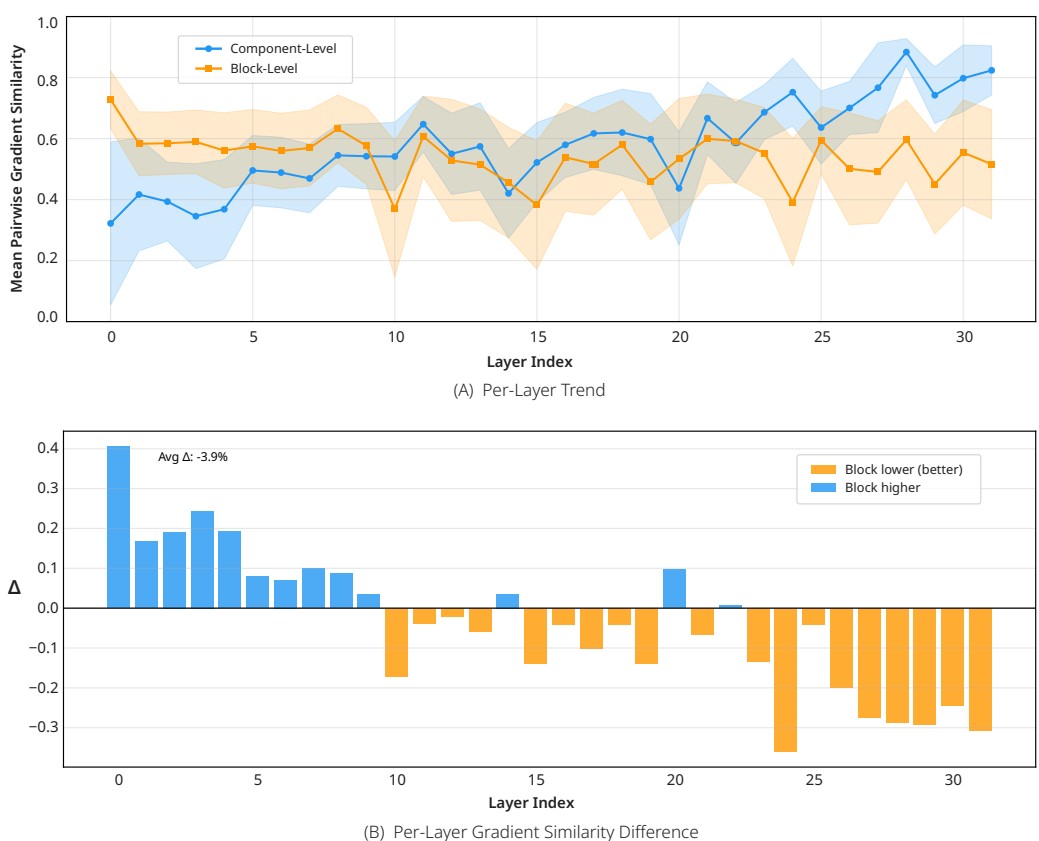

Figure S3: **Per-layer gradient similarity analysis.** **(A)** Per-layer mean pairwise gradient cosine similarity for component-level (red) and block-level (teal) adaptation. Error bands show standard deviation. **(B)** Per-layer difference (Block − Component). Negative values (teal) indicate block-level has lower gradient similarity (less conflict). Block-level shows up to 36% conflict reduction in later layers (24, 29, 31).

Table S4: **Domain difference in NLP vs vision.** Block-level adaptation improves both vision (DOTA: +2.2%, iNat2018: +1.6%) and NLP (Dolly-15k: +2.1%, BBH: +2.4%), while fine-grained routing shows mixed effects: it degrades DOTA (−1.3%) but improves iNat2018 (+0.3%) and NLP (Dolly-15k: +0.4%, BBH: +0.3%), indicating heterogeneous feature dimensions benefit from fine-grained routing. All results in accuracy (%)↑.

| Method | DOTA | iNat2018 | Dolly-15k | BBH | Avg. |
|---|---|---|---|---|---|
| Block-Level | | | | | |
|    HydraLoRA | 89.0 | 78.3 | 41.6 | 35.5 | 61.1 |
|    + Block-Level Adaptation | **91.2** | 79.9 | 43.7 | 37.9 | 63.2 |
| Routing | | | | | |
|    + Fine-Grained | 89.9 | **80.2** | **44.1** | **38.2** | **63.1** |

## C.4 DOMAIN DIFFERENCE ANALYSIS

Table S4 provides a detailed comparison of vision versus NLP domain performance (Section 4.5B). Block-level adaptation universally improves both vision (DOTA: +2.2%, iNat2018: +1.6%) and NLP (Dolly-15k: +2.1%, BBH: +2.4%). However, fine-grained routing shows mixed effects: it degrades DOTA (−1.3%) but improves iNat2018 (+0.3%) and NLP (+0.4% avg). This indicates that datasets with heterogeneous feature dimensions benefit from fine-grained routing, while homogeneous visual features may not require dimension-specific routing.

## C.5 BBH PERFORMANCE BREAKDOWN

Table S6 provides the detailed per-task performance on all 27 BBH tasks (Table 6 and Section 4.5A). mtLoRA achieves best results on 12 out of 27 tasks, with notable improvements on logical deduction tasks (*e.g.*, 7-object deduction: 28.3% vs 16.2% for HydraLoRA, +12.1%).

**Task Difficulty Categorization.** We categorize 27 BBH tasks by average accuracy across all methods into three difficulty levels, as shown in Table S5.

Table S5: BBH task categorization by difficulty level.

| Easy (>50%) | Medium (30-50%) | Hard (<30%) |
|---|---|---|
| formal fallacies | causal judgement | logical deduction 5 objects |
| boolean expressions | snarks | logical deduction 7 objects |
| movie recommendation | object counting | penguins in a table |
| sports understanding | logical deduction 3 objects | reasoning colored objects |
| hyperbaton | date understanding | salient translation error |
| navigate | tracking shuffled 3 | tracking shuffled 5 |
| web of lies | disambiguation qa | tracking shuffled 7 |
| | word sorting | dyck languages |
| | | geometric shapes |
| | | ruin names |
| | | temporal sequences |
| | | multistep arithmetic |
| *7 tasks* | *8 tasks* | *12 tasks* |

Table S6: **Per-task performance on BBH.** mtLoRA achieves best results on 12 out of 27 tasks. All results in accuracy (%)↑. Best per task in **bold**.

| Task | LoRA | MMoELoRA | HydraLoRA | mtLoRA |
|---|---|---|---|---|
| formal_fallacies | **100.00** | 95.55 | **100.00** | **100.00** |
| boolean_expressions | 72.47 | 70.04 | **94.74** | 71.66 |
| movie_recommendation | 61.94 | 69.64 | 36.84 | **84.21** |
| sports_understanding | 60.73 | 54.66 | **71.66** | 63.16 |
| hyperbaton | 48.18 | 48.18 | **64.37** | 59.51 |
| navigate | 56.28 | **57.89** | 57.49 | 57.49 |
| web_of_lies | **50.61** | 50.20 | **50.61** | **50.61** |
| causal_judgement | 48.37 | 48.91 | 47.83 | **51.63** |
| snarks | 45.71 | **46.86** | **46.86** | 45.71 |
| object_counting | 39.27 | 37.65 | **40.89** | 38.87 |
| logical_deduction_three_objects | 37.65 | 43.32 | 34.01 | **47.37** |
| date_understanding | 36.03 | **38.46** | 34.82 | 34.41 |
| tracking_shuffled_objects_3 | 32.79 | **36.44** | 35.63 | 34.82 |
| disambiguation_qa | 31.98 | 41.30 | 26.72 | **46.15** |
| word_sorting | 30.77 | **33.60** | **33.60** | 29.15 |
| logical_deduction_five_objects | 27.13 | 24.70 | 25.51 | **33.20** |
| penguins_in_a_table | 26.57 | 24.48 | 27.97 | **29.37** |
| logical_deduction_seven_objects | 20.24 | 17.81 | 16.19 | **28.34** |
| reasoning_about_colored_objects | 19.03 | 19.43 | 20.65 | **22.27** |
| salient_translation_error_detection | **17.81** | **17.81** | **17.81** | 17.00 |
| tracking_shuffled_objects_5 | 17.81 | **19.43** | 17.41 | 15.79 |
| tracking_shuffled_objects_7 | 11.74 | 10.93 | 10.12 | **15.79** |
| dyck_languages | 10.12 | 13.36 | **36.03** | 21.86 |
| geometric_shapes | 10.12 | 10.93 | **11.34** | 9.72 |
| ruin_names | 8.10 | 9.72 | **23.48** | 22.67 |
| temporal_sequences | 4.45 | 7.29 | **8.10** | 6.88 |
| multistep_arithmetic_two | 2.43 | **6.48** | 6.07 | 2.43 |
| **Average** | 34.38 | 35.37 | 36.92 | **38.52** |

## C.6 COMPUTATIONAL EFFICIENCY ANALYSIS

We provide detailed computational efficiency analysis in Tables S1-S2 (Section 4.5C), focusing on NLP benchmarks where all experiments were conducted on the same hardware ($2\times$ GPU DDP, LLaMA-2-7B).

**Main Ablation Efficiency.** Table S1 shows the parameter count and wall-clock training time. We have five key findings. 1) Block-level adaptation reduces training time by 33% (94.6 min $\rightarrow$ 63.0 min) while using 50% fewer parameters (75.5M $\rightarrow$ 37.7M). 2) Spectral-aware regularization adds minimal overhead (+5% time, no extra parameters). 3) Fine-grained routing adds modest overhead (+4% time, +5.6% parameters). 4) Full `mtLoRA` is still 24% faster than HydraLoRA baseline (72.1 min vs 94.6 min) with 47% fewer parameters. 5) FLOPs reduction is modest (0.85×-0.99×). This indicates that the wall-clock speedup primarily comes from improved GPU utilization (block-level avoids redundant routing computations at multiple positions), not FLOPs reduction.

**Block-Level Ablation Efficiency.** Table S2 shows the efficiency comparison for different block-level configurations. Key findings: 1) Block-level (Attn or FFN alone) uses 50% parameters (37.7M vs 75.5M) while achieving 33% faster training. 2) FFN-only is the most efficient: same parameters and time as Attn-only, but better performance (63.0% vs 62.3%). 3) Attn+FFN maintains efficiency advantage: 15% faster than component-level (80.3 min vs 94.6 min) despite same parameter count.

