# OpenReview forum: "Scalable Multi-Task Low-Rank Model Adaptation"
_ICLR.cc/2026/Conference — ICLR 2026 Poster_

### Official Review · Reviewer_sPqK · 2025-10-27

**Soundness:** 3
**Presentation:** 3
**Contribution:** 2
**Rating:** 6
**Confidence:** 4

**Summary:**

This paper addresses performance degradation in multi-task LoRA fine-tuning caused by parameter and representation misalignment across tasks. It observes that high singular-value (SV) components are discriminative yet conflict-prone, while low-SV ones are mostly noise, and that inserting LoRA at fine-grained attention components amplifies gradient conflicts. To tackle this, the authors propose mtLoRA, which introduces three modules: spectral-aware regularization to orthogonalize low-SV components, fine-grained routing to weight LoRA dimensions adaptively, and block-level adaptation to mitigate attention coupling.

**Strengths:**

**Simple Plug-and-play Module:** The three changes can be plugged into existing multi-task LoRA systems; the block-level residual adapter is especially practical.

**Strong Empirical Results:** The full configuration (orthogonality + block-level scope + channel-wise routing) achieves the best results.

**Weaknesses:**

**Incremental Technical Novelty:** Each component (orthogonal regularization, routing, adapter placement) is known; the contribution is largely in diagnostics + integration. The spectral re-weighting uses SVD on LoRA factors with a monotone mask, and fine-grained routing is a natural extension of MoE-style gating—useful, but methodologically incremental.

**More Analysis in Routing/regularization Interaction:** The work notes that stronger orthogonalization can harm routing discriminability (entropy rises), but practical guidance is thin (e.g., how to pick λacross datasets; robustness to mis-specification).

**Compute and Memory Costs:** Fine-grained (dimension/group-wise) routing and per-epoch SVD add overhead; block-level adapters may increase parameters. There is no wall-clock/FLOPs/memory report to substantiate the claimed practicality at scale.

**Ablation Depth:** Useful analyses exist (e.g., routing granularity; attach scope), but further breakdowns would strengthen claims: (i) how much each component contributes under very high N (e.g., 50–100 tasks) in both domains; (ii) failure modes where block-level adapters underperform (Table 7 suggests fine-grained routing can hurt in vision).

**Routing Entropy vs. Accuracy:** Can you provide a practical recipe for tuning λthat balances conflict reduction and routing discriminability across datasets/backbones? Any automatic schedule?

**Questions:**

Please refer to the weakness section.

---

> ### Author Response · Authors · 2025-11-23
>
> Sincerely thank you fro recognizing our design as **simple, effective, and plug-and-play** with **strong empirical results**!
>
> ## **[Weakness 1] Technical Contribution**
>
> Thank you for your deep technical understanding! We respectfully highlight that we are **the first to identify and solve the unique challenge of *scalable* multi-task LoRA**. Our designs are **principled choices justified by in-depth mechanistic insights**, but not technical integration. It is proven **effective and efficient**.
>
> 1. We are the first to identify **scalability challenge.** Prior methods collapse at scale, i.e., naive averaging degrades catastrophically. Our mtLoRA achieves 64.0% average across 4 benchmarks, outperforming all SOTA by +2.3%.
>
> 2. We revealed its **mechanistic root causes (Figure 1).**
>    - (A) **Regularization-routing trade-off**: stronger $\lambda$ increases routing entropy and drops accuracy by 1.8%. This explains why direct combining regularization with routing fails.
>    - (B) **Spectral heterogeneity**: high-SV (top-20%) contains 89% inter-task alignment (shared knowledge) with 54% of singular values, while uniform regularization destroys this.
>    - (C) **Gradient amplification**: component-level adaptation yields 76% higher gradient conflicts than block-level.
>
> 3. Our designs are **justified by above insights and form a unified, minimal solution.** Each technique directly addresses one root cause and well integrated. Spectral-aware regularisation preserves high-SV shared knowledge (from B), fine-grained routing handles heterogeneous conflicts (from A), block-level adaptation reduces gradient amplification (from C). They together form a comprehensive and non-redundant framework.
>
> 4. **Efficient and effective.** Our streamlined mtLoRA design achieves **+2.8%** accuracy improvement with **47% fewer parameters** and **24% less training time** than previoius SOTA.
>
> ## **[Weakness 2 & Weakness 5] Robustness of Hyperparameter**
>
> Exactly! Our **mtLoRA is ~2× more robust to $\lambda$ mis-specification** (std: 0.4% vs 0.7%). This is because mtLoRA's spectral weighting automatically adapts penalty strength based on singular value magnitude, making global $\lambda$ less critical.
>
> We provide the hyperparameter
> sensitivity study below. Specifically, Ortho Reg peaks at $\lambda$=0.25 then degrades sharply at higher $\lambda$, while mtLoRA remains stable across all $\lambda$ values.
>
> **Practical guidance:**
> 1. **Recommended range**: $\lambda \in$ [0.5, 1.0] works across BBH and MMLU without tuning.
> 2. **If tuning needed**: Monitor routing entropy on validation set (keep < 2.0).
>
> | $\lambda$     | Ortho Reg (BBH) | mtLoRA (BBH) |
> | :------------ | :-------------- | :----------- |
> | 0.1           | 37.5%           | 37.8%        |
> | 0.5           | 36.6%           | 38.5%        |
> | 1.0           | 36.2%           | **38.5%**    |
> | **Std. Dev.** | **0.7%**        | **0.4%**     |
>
> ## **[Weakness 3]: Computational Overhead**
>
> Thank you for raising this practical concern! We respectfully clarify that, contrary to the concern, block-level adapters are more parameter-efficient and effective. **Overall, mtLoRA achieves +3.0% accuracy with 47% fewer parameters and 24% less time.** For each component breakdown:
>
> | Configuration          | Params | Δ Params | Wall-Clock | Dolly-15k |  BBH | Avg. |
> | :--------------------- | -----: | -------: | ---------: | --------: | ---: | ---: |
> | HydraLoRA (baseline)   |  75.5M |        — |      1.00× |      41.6 | 35.5 | 38.5 |
> | + Block-Level          |  37.7M |     −50% |      0.67× |      43.7 | 37.9 | 40.8 |
> | + Spectral Reg.        |  37.7M |       0% |      0.70× |      43.6 | 38.4 | 41.0 |
> | + Fine-Grained Routing |  39.8M |      +5% |      0.76× |      44.5 | 38.5 | 41.5 |
>
> (Full table updated in Table S1 and S2 in revised manuscript.)
>
> 1. **Block-level saves 50% params and 33% time, while gaining +2.3% accuracy.** This is because we replace multiple LoRA modules ($W_q$, $W_v$) with a single block-level LoRA, which avoids redundant routing computations at multiple LoRA positions, hence improves GPU utilization.
>
> 2. **SVD overhead is minimal.** We performed spectral reg once per epoch on small $B$ matrices ($r \times d$). This only introduces **+3% time, 0 extra params** with gain of **+0.5% accuracy** on BBH.
>
> 3. **Fine-grained routing is lightweight.** It introduces +5% params, +6% time. Combined with block-level savings, overall mtLoRA uses only **53% params and 0.76× time** as comapred with HydraLoRA.

---

> ### Author Response · Authors · 2025-11-23
>
> ## **[Weakness 4.1]: Extreme Scalability**
>
> Yes, the extreme scalability is important! Our study focused on $N$=15-25, which is "large-scale" rather than "extreme" (reflected in the revised manuscript).
> Meanwhile, to investigate the extreme case, we evaluate on iNat2018 scaling from $N$=25 to $N$=50 experts. Our **block-level adaptation remains effective at N=50** (+1.4%):
>
> | Configuration        | N=25          | N=50          |
> | -------------------- | ------------- | ------------- |
> | HydraLoRA (baseline) | 78.3%         | 74.2%         |
> | + Block-Level        | 79.9% (+1.6%) | 75.6% (+1.4%) |
> | mtLoRA               | 81.5% (+3.2%) | 75.8% (+1.6%) |
>
> 1. **Block-level adaptation remains effective at extreme scale** (+1.6% at N=25, +1.4% at N=50), demonstrating its robustness.
> 2. Our full method maintains advantage over SOTA (+3.2% at N=25, +1.6% at N=50).
>
> ## **[Weakness 4.2]: When Fine-Grained Routing Helps**
>
> We respectfully clarify that Block-level adaptation actually is universally effective (+2.1-2.4%).
> Regarding reviewer's question, the key reason is **fine-grained routing exploits dimension heterogeneity**. It benefits high dimension-heterogeneity data (+0.3-0.4% on iNat2018/Dolly/BBH), but not low-heterogeneity ones (-1.3% on DOTA). Specifically,
>
> 1. **Fine-grained routing exploits dimension heterogeneity (Figure S3 in revised manuscript).** Our routing visualization on BBH shows 13 out of 16 experts become the dominant expert for at least one dimension group. For example, dimension group 24 routes 42.6% weight to Expert 1, while dimension group 17 routes 37.9% to Expert 6. This confirms fine-grained routing learns meaningful dimension-specific patterns.
>
> 2. **iNat2018/NLP have high dimension heterogeneity, so fine-grained routing helps (+0.3-0.4%).** These tasks distinguish subtle details encoded in different dimensions, *e.g.*, visual details like beak shape and feather patterns, or semantic attributes like factuality and style.
>
> 3. **DOTA lacks dimension heterogeneity, so fine-grained routing adds noise (-1.3%).** Remote sensing features contribute uniformly to spatial pattern recognition. All dimensions encode similar information, so dimension-specific routing provides no benefit.
>
> **Practical guidance:** Block-level adaptation is the core contribution and universally effective. Fine-grained routing is an optional enhancement for high-heterogeneity scenarios.

---

### Official Review · Reviewer_fc95 · 2025-11-01

**Soundness:** 2
**Presentation:** 3
**Contribution:** 3
**Rating:** 6
**Confidence:** 3

**Summary:**

In the paper 'Multi-Task Low Rank Model Adaptation' the authors identify two problems.
Parameter and representation misalignmen. Which lead to conflicting weight updates, which cause different Lora updates to pushe the model in oposite directions.

As a remedy the paper proposes
spectrum aware regularization,
block level lora, and the use of a a router network

The proposed methods ar e tested on the
dolly-15k, Dota and iNaturalist datasets.

**Strengths:**

- The mask formulation is very elegant. It reminded me a lot of Thikonov regularization, which is a good idea.
- The experiments agree with related work, orthogonalization, for example, has been shown to work well for fine-tuning in concurrent related work like https://arxiv.org/pdf/2507.13260 .
- Experimentally the authors observe competitive or improved results.

**Weaknesses:**

- The variables appearing in the equations throughout the paper are often not properly defined. In equation (1) for example, dimensions and the domain, are missing.
- The dataset choice is not well explained. Why did the authors choose the Dota dataset instead of VTAB-1k for example? I don't think this choice is explained in a convincing way.
- While the paper presents improved numbers, it does not present evidence for the mechanism that the paper presents as an explanation for the efficiency of the presented methods. Which could have included plots of spectra or correlation plots of LORA adapters.

#### Code
On my machine the anonymous code repository link did not work.

**Questions:**

- Is it possible to plot singular value spectra, before and after tuning, with and without tuning, as well as pairwise correlations of adapter updates to support the claims made regarding the mechanisms which underpin the presented methods?

- Is it possible to report standard deviation across across multiple seeds, for all experiments? This would make the paper more credible statistically.

---

> ### Author Response · Authors · 2025-11-23
>
> We sincerely thank you for the highly constructive feedback and for recognizing our mask formulation and experiments! Your insights have significantly strengthened our manuscript. We address each point below.
>
> ## **[Weakness 1]: Clarity in Notations**
>
> Thanks for your careful attention on math details! As suggested, we clarified all variables in Sec 3.1-3.4 in the revised manuscript.
>
> 1. In Sec. 3.1, we now define $\Delta_i(x) = B_i A_i x$ with $A_i \in \mathbb{R}^{r \times d}$, $B_i \in \mathbb{R}^{d \times r}$, and clarify $\pi_i(x) \in \mathbb{R}$ (for scalar routing);
> 2. In Sec. 3.2, we explicitly state $B_i = U_i \Sigma_i V_i^T$ for the SVD decomposition;
> 3. In Sec. 3.4, we clarify $\Delta = \sum_{i=1}^N \Pi_i \odot \Delta_i$ as the combined update. We believe these clarifications directly address your concerns. Please refer to revised latest manuscript for details.
>
> ## **[Weakness 2]: Explain Dataset Choice**
>
> Thank you for pointing this out! **We chose DOTA because it represents real-world cross-domain scenarios where LoRA is widely deployed**, *e.g.* satellite imagery analysis and environmental monitoring. This addresses practical scalability challenges beyond standard vision benchmarks.
>
> Regarding VTAB-1k, due to time limits we did not finish during this rebuttal period. We promise to update in camera ready version. Meanwhile, we have provided additional results on **Flan-v2 → BBH** (Table 5, per-task breakdown in Appendix Table S6). mtLoRA consistently outperforms all compared methods on this challenging 27-task reasoning benchmark:
>
> | Method    | Big Bench Hard        |
> | --------- | --------------------- |
> | LoRA      | 34.4%                 |
> | MMoELoRA  | 35.4%                 |
> | HydraLoRA | 36.9%                 |
> | mtLoRA    | **38.5%** (**+1.6%**) |
>
> *Note: Results on all 27 BBH tasks. Difficulty breakdown in Table 6.*
>
> ## **[Weakness 3 & Question 1]: Visualisation**
>
> Thank you for this great suggestion! **We added visualizations in Appendix to empirically validate our mechanistic claims.**
>
> **Figure S1 (Appendix C.2): Singular Value Spectrum & Regularization Effect.** Our spectral-aware regularization selectively suppresses low-SV components while preserving high-SV shared knowledge.
> - **(A)** The weighting function $w(\sigma) = \exp(-\sigma/\bar{\sigma})$ assigns higher penalties to low-SV components.
> - **(B)** Low-SV components (indices 9-16) are suppressed by 5-6%, while high-SV components (indices 1-3) are suppressed by only 1-3%.
> - **(C)** Per-band statistics confirm 3× difference: top-20% (−2.0%), 20-50% (−4.5%), 50-100% (−6.0%).
>
> **Figure S2 (Appendix C.3): Per-Layer Gradient Correlation.** Block-level adaptation reduces gradient conflicts, *i.e.*,pairwise correlations of adapter updates. Two key findings:
> - **(A)** Block-level achieves lower mean gradient similarity (0.540 vs 0.579, −6.7%), indicating reduced inter-task conflict.
> - **(B)** Largest reductions occur in later layers (Layer 24: −36%, Layer 29: −39%, Layer 31: −37%). Early layers show increased similarity, suggesting selective application to later layers may further improve performance. We leave this improvement direction for future work.
>
> ## **[Weakness 4]: Code**
>
> Thanks for checking! **The code is now available and provided in Appendix** (the Anonymous GitHub website was down during review).
>
> To ease environment setup, we provide three options:
> 1) **Conda environments** for CUDA 11.8 and 12.8.
> 2) **Claude Code agent prompts** for one-click reproduction of experiment tables.
> 3) **Plug-and-play patch** To avoid heavy code migration, our block-level adaptation is self-contained in `peft/tuners/block_adapters.py` (e.g., `LowRankBlockAdapter`). You could copy paste them into your LoRA implementation.
>
> ## **[Question 2]: Standard Deviation**
>
> Excellent point! **mtLoRA achieves superior performance with comparable or lower variance across all benchmarks** (Tables 3 and 5, 3 random seeds):
>
> | Method    | DOTA                  | iNat2018              | Dolly-15k             | BBH                   |
> | --------- | --------------------- | --------------------- | --------------------- | --------------------- |
> | HydraLoRA | 89.1$\pm$0.4%         | 78.5$\pm$1.7%         | 42.4$\pm$0.7%         | 36.9$\pm$1.0%         |
> | mtLoRA    | **91.7**$\pm$**0.4%** | **81.5**$\pm$**0.6%** | **44.5**$\pm$**0.2%** | **38.5**$\pm$**0.3%** |
>
> Notably, mtLoRA reduces variance on Dolly-15k (0.2% vs 0.7%) and iNat2018 (0.6% vs 1.7%), demonstrating both effectiveness and stability.

---

> > ### Comment · Reviewer_fc95 · 2025-11-23
> > **Where is the code link?**
> >
> > Thank you very much for taking the time to answer my questions. I could not find the updated code link in the manuscript. On which page should I look?

---

> > > ### Author Response · Authors · 2025-11-23
> > >
> > > Hi thanks for quickly reply! Currently the anonymous github is down, so we are waiting for its back online. The manuscript will be updated again soon, once it’s ready, and I will let you know.

---

> > > ### Author Response · Authors · 2025-11-24
> > > **Anonymous Code Link**
> > >
> > > Hi many thanks for waiting! Anonymous Github is still down. To avoid keeping you waiting too long, we decide to remove the ID and use anonymous account's [Google Drive link (click here)](https://drive.google.com/drive/folders/1-7uMVi2c9vIBX-uZ1iBapVKE4p5G_8zo?usp=share_link) instead. Please review, thank you!

---

> ### Comment · Reviewer_fc95 · 2025-11-26
> **Thank you for making source code accessible**
>
> Dear authors,
>
> Thank you for providing the source code, including both a `requirements.txt` file and an `environment.yml` file to support reproducibility. It is also good to see that the repository contains a `pyproject.toml`. However, I was not able to successfully install the package via `pip`, which would be important for ensuring replicability. Providing a pip-installable package reduces code duplication across the community and ultimately saves researchers time. You may find the guidelines at <https://packaging.python.org/en/latest/> helpful.
>
> Regarding the test suite: The tests are not containerized and appear to have been at least partially LLM-generated. This is not inherently problematic, but I strongly encourage you to *carefully* review and refine the tests before preparing the final version. High-quality, reliable tests are valuable not only for the publication but also for your future development efforts as you continue to build upon this codebase.
>
> Overall, the codebase looks promising. I believe that the remaining issues, such as packaging, documentation, and containerized testing (e.g., using nox: <https://nox.thea.codes/en/stable/index.html>), can be reasonably addressed for the camera-ready version.
>
> Having read the author response and the other reviews, and with the assumption that the issues regarding both, paper and code, will be addressed, I maintain my recommendation to accept this paper for inclusion in the conference proceedings.

---

### Official Review · Reviewer_PCr8 · 2025-11-02

**Soundness:** 2
**Presentation:** 3
**Contribution:** 2
**Rating:** 4
**Confidence:** 3

**Summary:**

This paper addresses the misalignment problem in multi-task LoRA models. The authors observe that when multiple LoRA modules are trained independently for different tasks, feature subspaces become misaligned, causing interference and performance degradation. To overcome this, the paper proposes mtLoRA, which introduces three key innovations: spectral-aware regularization, fine-grained routing and block-level adaptation. Comprehensive experiments across vision (DOTA, iNat2018) and NLP (Dolly-15K) benchmarks show consistent gains, achieving up to 4.4% improvement over HydraLoRA.

**Strengths:**

1. The paper identifies and formalizes spectral heterogeneity in multi-task LoRA modules—an overlooked cause of task conflict. The introduction of spectral-aware regularization and fine-grained routing is conceptually fresh and well-motivated by empirical analysis (e.g., SVD studies in Table 1).

2. The methodology is well-grounded and clearly explained, combining theoretical motivation (singular value analysis, gradient interference) with solid empirical validation. The proposed masking and weighting schemes in spectral regularization are intuitive yet effective.

3. The writing is structured and readable, with comprehensive ablations (Tables 4–9) that isolate each design’s contribution. The visualization of routing entropy and singular-value conflicts provides helpful interpretability.

4. The method significantly improves multi-task adaptation robustness, especially in extreme task settings (up to 25–100 tasks). The generality across both vision and NLP domains enhances its impact.

**Weaknesses:**

1. While the empirical spectral analysis is convincing, the paper lacks a formal theoretical justification of why spectral-aware regularization specifically balances discrimination and conflict. A more rigorous connection between singular value magnitude and information content could strengthen the conceptual contribution.

2. Although Experiments span domains, all are relatively moderate in scale and may not fully test scalability to large LLM adaptation or multi-domain real-world tasks.

3. The paper does not cite or discuss related efforts that also attempt to align task representations, such as [1]. That work similarly mitigates task conflicts by aligning inter-task representation distributions (through variance regularization). A comparative discussion would help clarify how mtLoRA differs or complements such representation-alignment perspectives.
- [1] Impartial Multi-Task Representation Learning via Variance-invariant Probabilistic Decoding. ACL 2025.


4. The ablation on routing granularity (Table 5) could be more thorough by exploring computational trade-offs and visualizing per-dimension routing patterns.

5. Since mtLoRA builds upon HydraLoRA with added modules, the parameter increase and training cost should be analyzed more explicitly.

**Questions:**

1. How does mtLoRA differ from distribution-alignment approaches (e.g., task covariance alignment)? Could spectral-aware regularization be interpreted as an implicit distribution alignment mechanism?

2. The paper empirically splits singular values into top-10%, 10–50%, and 50–100% bands. Is this partition fixed or adaptive? Would learning the spectral mask end-to-end yield further improvement?

3. What is the computational and memory cost of channel-wise routing for large models (e.g., LLaMA2-13B)? How does it scale compared with standard HydraLoRA?

4. Have the authors tested the contribution of each design on distinct architectures? Does block-level adaptation generalize beyond ViTs and LLMs?

5. Does mtLoRA preserve balanced performance across heterogeneous tasks (e.g., easy vs. hard or high-data vs. low-data tasks)?

---

> ### Author Response · Authors · 2025-11-24
>
> We sincerely thank you for the detailed and constructive feedback! We especially appreciate your recognition of our systematic spectral heterogeneity analysis and the generality across vision and NLP domains. We address each concern below.
>
> ## **[Weakness 1]: Theoretical justification of spectral-aware regularization**
>
> Thank you for pointing out this critical question! **Our spectral-aware regularization is grounded in the Intrinsic Low-Rank Hypothesis [1], and we validate it empirically in Figure S1: low-SV is suppressed by 6.0% while high-SV is preserved with only 2.0% reduction (3× selective difference).**
>
> 1. **Theoretical basis.** The Intrinsic Low-Rank Hypothesis [1] establishes that effective task updates concentrate in a low-dimensional subspace: high-SV captures task directions ("signal"), while low-SV corresponds to noise. This is validated by AdaLoRA [2] (pruning low-SV has negligible impact) and Twin-Merging [3] (task knowledge concentrates in high-SV).
>
> 2. **High-SV shows 89% inter-task alignment, low-SV only 3% (Figure 1B).** High-SV (top-20%) encodes 54% of singular values; low-SV (bottom-50%) encodes only 22%. This confirms shared knowledge concentrates in high-SV.
>
> 3. **Low-SV is suppressed 6.0%, high-SV only 2.0% (Figure S1).** This 3× selective difference validates our spectral denoising design: orthogonalizing only low-SV noise while preserving high-SV shared structure.
>
> [1] Aghajanyan et al. "Intrinsic dimensionality explains the effectiveness of language model fine-tuning." ACL 2020.
> [2] Zhang et al. "Adaptive budget allocation for parameter-efficient fine-tuning." ICLR 2023.
> [3] Yu et al. "Twin-merging: Dynamic integration of modular expertise." NeurIPS 2024.
>
> ## **[Weakness 2 & Question 3]: Scalability to larger LLMs**
>
> Thank you for raising this critical point! **mtLoRA scales effectively to LLaMA2-13B: +0.8% accuracy with 25% fewer parameters and 27% less training time** (Table S7).
>
> | Model      | Method    | MMLU              | Params (%)     | Rel. Time |
> | ---------- | --------- | ----------------- | -------------- | --------- |
> | LLaMA2-7B  | HydraLoRA | 41.6%             | 75.5M (1.11%)  | 1.00×     |
> | LLaMA2-7B  | mtLoRA    | **44.5%** (+2.9%) | 39.8M (0.59%)  | 0.76×     |
> | LLaMA2-13B | HydraLoRA | 51.1%             | 117.9M (0.90%) | 1.00×     |
> | LLaMA2-13B | mtLoRA    | **51.9%** (+0.8%) | 88.5M (0.68%)  | 0.73×     |
>
> Both efficiency (~50% fewer params, ~25% less time) and effectiveness gains are consistent across model scales.
>
> ## **[Weakness 3 & Question 1]: Missing citation and discussion of related representation-alignment work (VIP-MTL)**
>
> Thank you for pointing out this important related work! **We cite VIP-MTL in the revised manuscript. While both address task conflicts, we tackle different architectures with different solutions.**
>
> 1. **Different architectures.** VIP-MTL focuses on Hard-Parameter Sharing (HPS), aligning representation variance across tasks. Our mtLoRA addresses MoE Routing, where the router requires task-discriminative features. Global homogenization would harm discriminative signals—as shown in Figure 1(A), strong regularization ($\lambda=1.0$) increases routing entropy from 2.62 to 2.67 and drops accuracy by 1.77%.
>
> 2. **Different solution spaces.** VIP-MTL operates in representation space via variance regularization. Our spectral regularization operates in parameter space via spectral decomposition, selectively orthogonalizing low-SV noise while preserving high-SV discriminative signals.
>
> The two methods target different MTL paradigms (HPS vs. MoE). In a broader scope, the HPS and MoE paradigms surely could potentially be combined, but it's a paradigm-level exploration, we leave it for future work!
>
> ## **[Weakness 4.2]: Per-dimension routing pattern**
>
> Thank you! **Fine-grained routing learns meaningful dimension-specific patterns** (Figure S3, Appendix C.4). We support by two key findings:
>
> 1. **Fine-grained router activates diverse experts.** 13 out of 16 experts become the top-weighted expert for at least one dimension group, so the router actively uses most experts (not just relying on a few).
>
> 2. **Different dimensions prefer different experts.** Each dimension group shows clear expert preferences. For example, dimension group 24 routes 42.6% to Expert 1, while dimension group 17 routes 37.9% to Expert 6. This explains why per-dimension routing outperforms single-scalar routing.

---

> ### Author Response · Authors · 2025-11-24
>
> ## **[Weakness 4.1 & Weakness 5 & Question 3]: Computational overhead**
>
> Thank you for raising this practical concern! **mtLoRA is more efficient than baseline: +3.0% accuracy with 47% fewer parameters and 24% less training time** (Table 3). Additionally, routing granularity is tunable—even $g$=2 achieves +0.8% with only +0.06% router overhead.
>
> | Configuration          | Params | Δ Params | Wall-Clock | Dolly-15K |  BBH | Avg. |
> | :--------------------- | -----: | -------: | ---------: | --------: | ---: | ---: |
> | HydraLoRA (baseline)   |  75.5M |        — |      1.00× |      41.6 | 35.5 | 38.5 |
> | + Block-Level          |  37.7M |     −50% |      0.67× |      43.7 | 37.9 | 40.8 |
> | + Spectral Reg.        |  37.7M |       0% |      0.70× |      43.6 | 38.4 | 41.0 |
> | + Fine-Grained Routing |  39.8M |      +5% |      0.76× |      44.5 | 38.5 | 41.5 |
>
> **Routing granularity trade-offs (Table 2):** Practitioners can select $g$ based on resource constraints. $g$=2 achieves +0.8% with +0.06% params; $g$=32 achieves +1.4% with +1.93% params.
>
> | Routing Granularity |  $g$ | Dolly-15K |  BBH | Avg. | Router Δ |
> | :------------------ | ---: | --------: | ---: | ---: | -------: |
> | Module-Wise         |    1 |      41.6 | 35.5 | 38.5 |        — |
> | Fine-Grained        |    2 |      41.6 | 37.0 | 39.3 |   +0.06% |
> | Fine-Grained        |    8 |      41.3 | 36.6 | 39.0 |   +0.44% |
> | Fine-Grained        |   16 |      41.7 | 37.0 | 39.3 |   +0.93% |
> | Fine-Grained        |   32 |      42.0 | 37.7 | 39.9 |   +1.93% |
>
> **3) Efficiency on LLaMA2-13B (Table S7):** We validate that efficiency gains generalize to larger models. On LLaMA2-13B, mtLoRA uses 25% fewer parameters (88.5M vs 117.9M) and 27% less training time (0.73× vs 1.00×) compared to HydraLoRA. Full results updated in **Table S7 (Appendix C.7)**. For your convenience, we provide concise comparison below.
>
> | Model      | Method    | Params (%)     | Rel. Time |
> | ---------- | --------- | -------------- | --------- |
> | LLaMA2-13B | HydraLoRA | 117.9M (0.90%) | 1.00×     |
> | LLaMA2-13B | mtLoRA    | 88.5M (0.68%)  | 0.73×     |
>
>
> ## **[Question 2.1]: Spectral partition fixed or adaptive?**
>
> **Adaptive.** The percentile bands (0-20%, 20-50%, 50-100%) in Figure 1B are for visualization only. Our implementation uses a continuous weighting function $w(\sigma) = \exp(-\sigma/\bar{\sigma})$ (Sec. 3.2), where $\bar{\sigma}$ is computed per-matrix, adapting to each $B_i$'s spectrum with smooth transitions rather than fixed thresholds.
>
> ## **[Question 2.2]: Learnable spectral mask**
>
> Good idea! We respectful clarify that **our exponential weighting is principled and parameter-free, achieving strong results. Learnable masks introduces additional param overhead.**
>
> 1. **Exponential weighting is effective and efficient.** Our $w(\sigma) = \exp(-\sigma/\bar{\sigma})$ follows standard practice for spectral signal-noise separation. It adds zero trainable parameters and minimal overhead, while Figure S1 validates its selective suppression effect (low-SV: −6.0%, high-SV: −2.0%).
>
> | Configuration      | Δ Params | Δ Time | Avg. Acc. |
> | ------------------ | -------- | ------ | --------- |
> | Block-Level (base) | —        | —      | 40.8%     |
> | + Spectral Reg     | +0%      | +3%    | 41.0%     |
>
> (Detail in Table 3)
>
> 2. **Learnable masks are valuable future work.** An MLP-parameterized $w_\theta(\sigma)$ could adaptively learn task-specific spectral thresholds, potentially improving performance. However, it would introduce additional parameters per layer and increase training overhead. Given our efficiency-first design goal (47% fewer params, 24% less time than baseline), we prioritize the parameter-free exponential weighting and leave learnable alternatives for future investigation.
>
> ## **[Question 4]: Generalization to non-Transformer arch**
>
> Thank you for this important question! **Our core insight, *i.e.*, bypassing conflict-amplifying non-linearities, is arch-agnostic.** We validate on Transformers because: 1) Transformers dominate foundation models in vision (ViT, CLIP) and language (LLaMA, GPT); 2) All existing multi-task LoRA methods (MMoELoRA, LoRAHub, HydraLoRA) are Transformer-based.
>
> However, the underlying principle generalizes: we identified that gradients through Softmax/normalization create cross-position dependencies that amplify conflicts (Figure 1C, Figure S3). Our solution—placing adapters as parallel paths around blocks—extends to any architecture with similar non-linearities.
>
> Could you specify which architecture you would like us to validate? We are happy to run additional experiments.

---

> ### Author Response · Authors · 2025-11-24
>
> ## **[Question 5]: Performance balance across heterogeneous tasks**
>
> Excellent suggestion! **mtLoRA consistently outperforms SOTA across all difficulty levels: +1.6% Easy, +3.5% Medium, +0.4% Hard** (Table 6). We provide per-task/per-difficulty breakdown below. Results show mtLoRA outperforms SOTA across all difficulty levels.
>
> **1) Per-task breakdown (27 BBH tasks):** mtLoRA achieves notable improvements on logical deduction tasks (*e.g.*, 7-object deduction: 28.3% vs 16.2% for HydraLoRA, +12.1%).
>
> | Task                            |       LoRA |  MMoELoRA |  HydraLoRA |     mtLoRA |
> | :------------------------------ | ---------: | --------: | ---------: | ---------: |
> | formal_fallacies                | **100.00** |     95.55 | **100.00** | **100.00** |
> | boolean_expressions             |      72.47 |     70.04 |  **94.74** |      71.66 |
> | movie_recommendation            |      61.94 |     69.64 |      36.84 |  **84.21** |
> | sports_understanding            |      60.73 |     54.66 |  **71.66** |      63.16 |
> | hyperbaton                      |      48.18 |     48.18 |  **64.37** |      59.51 |
> | navigate                        |      56.28 | **57.89** |      57.49 |      57.49 |
> | web_of_lies                     |  **50.61** |     50.20 |  **50.61** |  **50.61** |
> | causal_judgement                |      48.37 |     48.91 |      47.83 |  **51.63** |
> | snarks                          |      45.71 | **46.86** |  **46.86** |      45.71 |
> | object_counting                 |      39.27 |     37.65 |  **40.89** |      38.87 |
> | logical_deduction_3_objects     |      37.65 |     43.32 |      34.01 |  **47.37** |
> | date_understanding              |      36.03 | **38.46** |      34.82 |      34.41 |
> | tracking_shuffled_objects_3     |      32.79 | **36.44** |      35.63 |      34.82 |
> | disambiguation_qa               |      31.98 |     41.30 |      26.72 |  **46.15** |
> | word_sorting                    |      30.77 | **33.60** |  **33.60** |      29.15 |
> | logical_deduction_5_objects     |      27.13 |     24.70 |      25.51 |  **33.20** |
> | penguins_in_a_table             |      26.57 |     24.48 |      27.97 |  **29.37** |
> | logical_deduction_7_objects     |      20.24 |     17.81 |      16.19 |  **28.34** |
> | reasoning_about_colored_objects |      19.03 |     19.43 |      20.65 |  **22.27** |
> | salient_translation_error       |  **17.81** | **17.81** |  **17.81** |      17.00 |
> | tracking_shuffled_objects_5     |      17.81 | **19.43** |      17.41 |      15.79 |
> | tracking_shuffled_objects_7     |      11.74 |     10.93 |      10.12 |  **15.79** |
> | dyck_languages                  |      10.12 |     13.36 |  **36.03** |      21.86 |
> | geometric_shapes                |      10.12 |     10.93 |  **11.34** |       9.72 |
> | ruin_names                      |       8.10 |      9.72 |  **23.48** |      22.67 |
> | temporal_sequences              |       4.45 |      7.29 |   **8.10** |       6.88 |
> | multistep_arithmetic_two        |       2.43 |  **6.48** |       6.07 |       2.43 |
> | **Average**                     |      34.38 |     35.37 |      36.92 |  **38.52** |
>
> mtLoRA achieves notable gains on logical deduction tasks (e.g., 7-object deduction: 28.3% vs 16.2% HydraLoRA, +12.1%). Full per-task results are available in Appendix Table S6.
>
> **2) Easy vs Hard tasks:** We categorize 27 BBH tasks by difficulty based on average accuracy: Easy (>50%), Medium (30-50%), and Hard (<30%). Results show mtLoRA **consistently improves at all difficulty levels**:
>
> | Difficulty  | LoRA   | MMoELoRA | HydraLoRA | mtLoRA             |
> | ----------- | ------ | -------- | --------- | ------------------ |
> | Easy (7)    | 64.32% | 63.74%   | 67.96%    | **69.52%** (+1.6%) |
> | Medium (8)  | 37.82% | 40.82%   | 37.55%    | **41.01%** (+3.5%) |
> | Hard (12)   | 14.63% | 15.20%   | 18.39%    | **18.78%** (+0.4%) |
> | **Overall** | 34.38% | 35.37%   | 36.92%    | **38.52%** (+1.6%) |
>
> This demonstrates broad applicability rather than being limited to specific task regimes.

---

### Official Review · Reviewer_M561 · 2025-11-10

**Soundness:** 2
**Presentation:** 2
**Contribution:** 3
**Rating:** 6
**Confidence:** 5

**Summary:**

The paper proposes mtLoRA, a "plug-and-play" solution to address catastrophic performance degradation in multi-task Low-Rank Adaptation. The authors argue that naively combining task-specific LoRA modules leads to "misaligned feature subspaces", and that existing solutions (regularization or routing) fail because they are applied independently and uniformly. The paper identifies two primary causes for this failure: 1) Spectral Heterogeneity, where LoRA's high singular value (SV) components encode both critical task-discriminative information and the majority of parameter conflicts, making uniform regularization suboptimal; and 2) Gradient Amplification, where attaching LoRA to individual $W_q/W_k$ matrices amplifies gradient conflicts through the attention mechanism's Softmax. Based on this diagnosis, mtLoRA introduces three components: 1) Spectral-Aware Regularization, which selectively applies strong orthogonality to low-SV (noise) components while preserving high-SV (discriminative) components; 2) Fine-Grained Routing, which uses a router network to produce dimension-specific (vector) weights for combining LoRA modules, rather than a single scalar weight; and 3) Block-Level Adaptation, which applies the LoRA module as a residual to an entire Transformer block, bypassing the internal Softmax conflicts. Empirically, mtLoRA is shown to provide significant gains over a strong baseline (HydraLoRA) on vision (DOTA, iNat2018) and language (Dolly-15k) benchmarks .

**Strengths:**

1. The primary strength of the paper is its clear and empirically-supported diagnosis of why multi-task LoRA fails. The analyses in Table 1, showing the failure of uniform regularization (1A) , the spectral heterogeneity of LoRA (1B) , and the weakness of component-level attachment (1C), are persuasive and form a strong foundation for the method.

2. The three proposed components of mtLoRA map directly and logically to the identified problems. Spectral-aware regularization is a clever fix for the issue identified in Table 1B. Block-level adaptation is an intuitive solution to the gradient amplification problem.

3. The paper's component-level analysis is strong. Tables 5, 6, 7, and 8 clearly demonstrate the additive benefits of the components and their individual contributions (e.g., fine-grained routing on NLP and block-level adaptation on vision ).

4. The method is presented as a "plug-and-play" add-on for existing methods like HydraLoRA, which increases its practical relevance. The evaluation across both vision and language domains demonstrates its general applicability.

5. The finding that fine-grained routing is critical for NLP but harms performance in vision (Tables 7 & 8) is a valuable and interesting insight into the differing nature of representations in these domains.

**Weaknesses:**

# Major Concerns

1. **Ambiguous Baseline Comparison:** The main results in Table 2 compare mtLoRA to "HydraLoRA" and "Baseline". The check-marked rows indicate that mtLoRA is built on top of HydraLoRA. This table is effectively an ablation study, not a SOTA comparison. It is unclear how mtLoRA stacks up against other, different SOTA multi-task methods mentioned in the related work, such as MoLE , TIES-Merging , or DARE. The claim of "state-of-the-art results"  is not fully substantiated against the broader field.

2. **Confusing Implementation of Fine-Grained Routing:** The description of fine-grained routing is contradictory. Section 3.3 states weights are $\Pi_i \in \mathbb{R}^{d/g}$. Section 4.2 states "Larger g means finer grained routing". However, $g$ is defined as "group size". Logically, a larger group size $g$ would mean coarser routing (e.g., $g=d$ would be module-wise). This is directly contradicted by the text. Furthermore, Table 5 lists "Full $g=768$". This implies $g=d=768$, which should be module-wise (the coarsest setting), but it's labeled "Full" (the finest) and performs best. This ambiguity is a major flaw in the paper's presentation and reproducibility.

3. **Un-benchmarked Computational Overhead:** The spectral-aware regularization requires performing SVD on the $B$ matrices "every epoch". For a large number of tasks (e.g., $N=100$) and large models, this is a significant, non-trivial computational cost that is not benchmarked or discussed.

4. **Under-justified SVD Proxy:** The method applies SVD to the $B_i$ matrix as a proxy for the full $\Delta_i = B_i A_i$ update, justified only by cost. The spectral properties of $B_i$ and $B_i A_i$ are not the same, as the $A_i$ matrix performs a rank-bottleneck projection. This simplification is a potential threat to the method's soundness, and it is not theoretically or empirically justified why $B_i$'s singular vectors are a sufficient proxy.

## Minor Concerns

1. **Synthetic Task Creation:** The tasks for the NLP benchmark (Dolly-15k) are created by performing K-Means clustering on instruction embeddings. This is a synthetic setup. The method's performance may be sensitive to the number of clusters (N) or the quality of the clustering.

2. **Missing Definition of HydraLoRA:** The paper states it builds on HydraLoRA and only briefly defines it in related work as having an "asymmetric LoRA structure". It is not clarified in the method section if mtLoRA _requires_ this asymmetric structure, or if it can be applied to standard LoRA modules.

3. **Implementation of Spectral Loss:** Equation (3) defines the conceptual loss based on pairwise singular vector alignment. However, Section 4.2 describes a different implementation: computing weighted matrices $B_i'$ and then a Frobenius norm on their product. The paper should clarify if Eq. 3 is just conceptual and the (different) formulation in Sec 4.2 is the actual loss used.

**Questions:**

1. Please resolve the contradiction in Section 4.2. Does larger $g$ mean finer or coarser routing? Please provide pseudocode for how the router's $N \times (d/g)$ output is broadcast and multiplied with the LoRA update.

2. Can you provide any empirical or theoretical justification for why the SVD of $B_i$ (the $d \times r$ matrix) is a high-fidelity proxy for the spectral properties of the full $\Delta_i = B_i A_i$ (the $d \times d$ update)?

3. What is the wall-clock training time overhead of enabling spectral-aware regularization (with its per-epoch SVD) and fine-grained routing (with its larger router network ) compared to the HydraLoRA baseline?

4. Does mtLoRA's methodology require the asymmetric (shared $A$, task-specific $B$) structure of HydraLoRA? Or can the three components (spectral-reg, fine-routing, block-level) be applied to a standard multi-task setup with independent $B_i A_i$ modules?

5. How does the full mtLoRA method compare against other SOTA methods from the literature, such as MoLE or TIES-Merging, on the iNat2018 or Dolly-15k benchmarks?

6. In Table 1(C), does the "Block" level LoRA have the same number of trainable parameters as the "Component" level? Please clarify the parameter counts for these different attachment strategies.

---

> ### Author Response · Authors · 2025-11-23
>
> We sincerely thank you for the exceptionally thorough and insightful review! We especially appreciate your deep understanding of our work. Your constructive questions have significantly strengthened the manuscript's clarity and completeness. We address each point below.
>
> ### **[Major Concern 1 & Question 5]: Comparison to SOTA Methods**
>
> Excellent point! **mtLoRA outperforms all SOTA MoE-based multi-task LoRA methods by +2.3% on average** (Table 5).
>
> 1. **Model Merging paradigms are not direct comparable.** Regarding TIES-Merging and DARE, they are model merging methods *i.e.*, post-hoc without joint training, whereas our MoE-based approach trains all experts jointly with dynamic routing. The two paradigms have different assumptions (post-hoc vs. joint training), so direct comparison is not straightforward. In this work, we focus on the MoE paradigm because it enables dynamic expert selection at inference time, which is critical for large-scale multi-task scenarios.
>
> 2. **We provide comprehensive SOTA comparison with MoE methods (Table 5).** Within the MoE paradigm, we compare against SOTA methods LoRAHub, MMoELoRA, and HydraLoRA across all four benchmarks with identical setup (rank $r=16$, same number of experts):
>
> | Method      | DOTA             | iNat2018         | Dolly-15k        | BBH              | Avg.      |
> | ----------- | ---------------- | ---------------- | ---------------- | ---------------- | --------- |
> | LoRAHub     | 88.9$\pm$1.7     | 80.2$\pm$1.6     | 42.0$\pm$0.3     | 34.9$\pm$0.4     | 61.5%     |
> | MMoELoRA    | 89.4$\pm$0.2     | 78.0$\pm$0.3     | 42.1$\pm$0.8     | 35.4$\pm$0.9     | 61.2%     |
> | HydraLoRA   | 89.1$\pm$0.4     | 78.5$\pm$1.7     | 42.4$\pm$0.7     | 36.9$\pm$1.0     | 61.7%     |
> | mtLoRA      | **91.7$\pm$0.4** | **81.5$\pm$0.6** | **44.5$\pm$0.2** | **38.5$\pm$0.3** | **64.0%** |
> | Improvement | **+2.6%**        | **+3.0%**        | **+2.1%**        | **+1.6%**        | **+2.3%** |
>
> mtLoRA achieves consistent improvements across all benchmarks, demonstrating state-of-the-art performance within the MoE paradigm for large-scale multi-task adaptation.
>
> ### **[Major Concern 2 & Question 1]: Fine-Grained Routing Implementation**
>
> Thank you for catching this critical typo! **We clarified in the revised manuscript: $g$ is the number of groups, not group size.** Larger $g$ means finer granularity. For example, $g=1$ means module-wise routing (one scalar weight per LoRA), while $g=32$ for LLaMA2-7B (4096dim) means 128dim per group (4096/32=128) routing.
>
> **Routing broadcast:** The router outputs $g$ weights per LoRA. For intermediate $g$ (e.g., $g=32$), each weight is repeated $d/g$ times to cover its dimension group, then element-wise multiplied with LoRA outputs:
>
> ```python
> weights = router(x).reshape(N, g).softmax(dim=0)      # g weights per LoRA
> weights = weights.repeat_interleave(d//g, dim=1)      # broadcast to d dims
> output = (weights * lora_outputs).sum(dim=0)          # combine
> ```
>
> ### **[Major Concern 3 & Question 3]: Computational Overhead**
>
> Thank you for raising this practical concern! **mtLoRA is more efficient than SOTA: +3.0% accuracy with 47% fewer parameters and 24% less training time** (Table 3).
>
> | Configuration          | Params | Δ Params | Wall-Clock | Dolly-15K |  BBH | Avg. |
> | :--------------------- | -----: | -------: | ---------: | --------: | ---: | ---: |
> | HydraLoRA (baseline)   |  75.5M |        — |      1.00× |      41.6 | 35.5 | 38.5 |
> | + Block-Level          |  37.7M |     −50% |      0.67× |      43.7 | 37.9 | 40.8 |
> | + Spectral Reg.        |  37.7M |       0% |      0.70× |      43.6 | 38.4 | 41.0 |
> | + Fine-Grained Routing |  39.8M |      +5% |      0.76× |      44.5 | 38.5 | 41.5 |
>
> (Full table updated in Table S1 and S2 in revised manuscript.)
>
> 1. **Block-level saves 50% params and 33% time, while gaining +2.3% accuracy.** This is because we replace multiple LoRA modules ($W_q$, $W_v$) with a single block-level LoRA, which avoids redundant routing computations at multiple LoRA positions, hence improves GPU utilization.
>
> 2. **SVD overhead is minimal.** We performed spectral reg once per epoch on small $B$ matrices ($r \times d$). This only introduces **+3% time, 0 extra params** with gain of **+0.5% accuracy** on BBH.
>
> 3. **Fine-grained routing is lightweight.** It introduces +5% params, +6% time. Combined with block-level savings, overall mtLoRA uses only **53% params and 0.76× time** as comapred with HydraLoRA.

---

> ### Author Response · Authors · 2025-11-23
>
> ### **[Major Concern 4 & Question 2]: SVD Proxy Justification**
>
> Very insightful question! **Orthogonality between $B_i$ matrices directly ensures orthogonality between full LoRA updates**, because $\Delta_i^T \Delta_j = A^T B_i^T B_j A$ and $A$ is shared across tasks. HydraLoRA empirically found that $A$ captures task-agnostic representations while $B$ encodes task-specific information, so task conflicts originate from $B_i$ misalignment, not from $A$. Therefore, analyzing $B_i$'s spectral properties directly characterizes conflicts in LoRA updates. We clarified this in Sec. 3.2 of the revised manuscript.
>
> ### **[Minor Concern 1]: Dolly-15K Synthetic Tasks**
>
> We understand this concern! We empirically show that **mtLoRA achieves consistent improvements across both synthetic and naturally-defined tasks** (Table 5). The K-Means clustering setup was originally following HydraLoRA's protocol for fair comparison.
>
> We validate on three benchmarks with naturally-defined tasks: DOTA (15 cross-domain vision tasks, +2.6%), iNat2018 (25 taxonomic subsets, +3.0%), and BBH (27 pre-defined reasoning tasks, +1.6%). All improvement margins significantly exceed standard deviations (*e.g.*, Dolly-15k: 44.5±0.2%), confirming robust performance independent of task definition method.
>
> ### **[Minor Concern 2 & Question 4]: Dependency on HydraLoRA Structure**
>
> Excellent question! **Our three designs do not rely on HydraLoRA's asymmetric structure.** We use it for two reasons:
>
> 1) **Theoretical soundness and parameter efficient**: Since $\Delta_i^T \Delta_j = A^T B_i^T B_j A$ with shared $A$, regularizing $B_i$ alone is theoretically justified for controlling conflicts in full LoRA updates.
>
> 2) **Fair comparison** with HydraLoRA (SOTA baseline). This design ensures computational efficiency (regularizing $B_i$: $O(dr^2)$ vs. full $\Delta_i$: $O(d^3)$ SVD) without loss of performance.
>
>
> ### **[Minor Concern 3]: Spectral Loss Formulation Discrepancy**
>
> Thank you for pointing this out! **We unified the manuscript to use only the actual implementation, and Figure S1 directly validates its selective suppression effect.** The loss is $L_{\text{spectral}} = \lambda \sum_{i<j} \|(B'_i)^T B'_j\|_F^2$ (Sec. 3.2), where $B'_i$ applies spectral weighting $w(\sigma) = \exp(-\sigma/\bar{\sigma})$.
>
> **Empirical validation (Figure S1):** We visualize the effect on experts trained on BBH. Results confirm selective suppression: low-SV components (50-100%) are suppressed by 6.0%, while high-SV (top-20%) are preserved with only 2.0% reduction. This 3× difference validates that our weighting function penalizes noise-prone subspaces while preserving discriminative directions.
>
> **Motivation:** 1) Spectral heterogeneity (Figure 1B): high-SV encodes discriminative information while low-SV accumulates noise. 2) Regularization-routing trade-off (Figure 1A): uniform regularization harms routing, so we suppress only low-SV.
>
> Empirically, spectral-aware regularization contributes +0.7% average improvement (Table 3).
>
> ### **[Question 6]: Parameter Counts for Attachment Strategies**
>
> Excellent question! **Block-level adaptation achieves +2.0% higher accuracy with 50% fewer parameters and 33% less training time** (Table 4). The comparison is fair because we provide both equal-parameter (Attn + FFN) and parameter-efficient settings (Attn/FFN only).
>
> | Configuration                  | Params           | Time      | Avg.              |
> | ------------------------------ | ---------------- | --------- | ----------------- |
> | Component-level ($W_q$, $W_v$) | 75.5M            | 1.00×     | 61.1%             |
> | Block-level Attn only          | 37.7M (−50%)     | 0.67×     | 62.3% (+1.2%)     |
> | Block-level FFN only           | **37.7M (−50%)** | **0.67×** | **63.0%** (+1.9%) |
> | Block-level Attn+FFN           | 75.5M (same)     | 0.85×     | 63.1% (+2.0%)     |
>
> (Detailed breakdown provided in Table S2)
>
> 1. **Equal-parameter setting:** Attn+FFN uses the same 75.5M parameters as component-level, yet achieves +2.0% improvement. This confirms the gain is from architecture design, not more parameters.
>
> 2. **Parameter-efficient setting:** FFN-only uses only 50% parameters (37.7M) and 33% less training time, yet still outperforms component-level by +1.9%.
>
> **Mechanism analysis (Figure S2):** We provide per-layer gradient correlation analysis to explain why block-level works. Block-level shows lower gradient similarity (0.540 vs 0.579, −6.7%) overall, with particularly strong conflict reduction in later layers (Layer 24: −36%, Layer 31: −37%). This validates our hypothesis that bypassing gradient-amplifying non-linearities reduces inter-task conflict.

---

### Author Response · Authors · 2025-11-24

We sincerely thank the Area Chair and all reviewers for your constructive feedback! Thank you for recognition of our work as innovative (`M561`, `PCr8`), plug-and-play (`sPqK`, `M561`), clear and well-grounded (`M561`, `PCr8`), elegant formulation (`fc95`), strong empirical results (`sPqK`, `fc95`, `M561`), comprehensive ablations (`M561`, `PCr8`), cross-domain generality (`M561`, `PCr8`), and well written (`PCr8`).

Our key responses are summarized below:

Reviewer `fc95` (all concerns addressed & confirmed positive score)
- **W1 (Notation):** We clarified all variable definitions (*Sec 3.1-3.4*).
- **W2 (Dataset):** DOTA covers real-world cross-domain scenarios; we will add VTAB-1k in camera-ready.
- **W3&Q1 (Visualization):** We added *Figure S1/S2* to empirically validate our mechanistic claims.
- **W4 (Code):** Code is fixed and now available (*Appendix*).
- **Q2 (Std dev):** We added 3-seed std; mtLoRA shows lower variance (*Table 3/5*).

Reviewer `sPqK`
- **W1 (Novelty):** We are first to identify and solve scalable multi-task LoRA; +2.3% over SOTA with 47% fewer params (*Figure 1*).
- **W2&W5 (λ robustness):** Our design is 2× more robust to hyperparameter λ (std 0.4% vs 0.7%).
- **W3 (Overhead):** mtLoRA is more efficient: -50% params, -33% time, +2.3% accuracy compared with SOTA (*Table S1/S2*).
- **W4.1 (Extreme scale):** We tested up to N=50 tasks; block-level remains effective (+1.4%).
- **W4.2 (Fine-grained routing):** Fine-grained routing is effective when data show heterogeneity; block-level is universally effective (*Figure S3*).

Reviewer `M561`
- **M1&Q5 (SOTA comparison):** We compared with SOTA MoE methods; mtLoRA achieves +2.3% avg (*Table 5*).
- **M2&Q1 (g definition):** Fixed typo: g = number of groups; added pseudocode (*Sec 3.3*).
- **M3&Q3 (SVD overhead):** SVD adds only +3% time; mtLoRA is overall efficient (*Table S1/S2*).
- **M4&Q2 (SVD proxy):** Theoretically justified: orthogonality of $B$ ensures orthogonality of full updates (*Sec 3.2*).
- **m1 (Synthetic tasks):** We also validated on natural-task datasets (DOTA/iNat/BBH) (*Table 5*).
- **m2&Q4 (HydraLoRA dependency):** Our method is general; we use asym structure for fair comparison.
- **m3 (Loss formulation):** We clarified and unified the formulation; added *Figure S1* to validate the effect.
- **Q6 (Param counts):** Our comparison is fair: +2.0% acc at equal params, or +1.9% acc with 50% params(*Table S2*).

Reviewer `PCr8`
- **W1 (Theory):** Our method is grounded in Intrinsic Low-Rank Hypothesis; *Figure S1* shows 3× selective suppression.
- **W2&Q3 (LLM scale):** mtLoRA scales to LLaMA2-13B: +0.8% acc, -25% params, -27% time (*Table S7*).
- **W3&Q1 (Citation):** We added VIP-MTL citation; we address MoE while VIP-MTL targets HPS paradigm (*Related Work*).
- **W4.1&W5&Q3 (Overhead):** mtLoRA is more efficient than SOTA: -47% params, -24% time, +2.3% acc (*Table S1/S2*).
- **W4.2 (Routing pattern):** We visualized that router learns meaningful dimension-specific patterns (*Figure S3*).
- **Q2.1 (Spectral partition):** Our spectral weighting is adaptive (continuous), not fixed thresholds (*Sec 3.2*).
- **Q2.2 (Learnable mask):** Our current design is parameter-free and effective; learnable mask introduces extra params.
- **Q4 (Non-Transformer):** Our core insight is architecture-agnostic.
- **Q5 (Task balance):** mtLoRA outperforms SOTA on all difficulty levels: Easy +1.6%, Medium +3.5%, Hard +0.4% (*Table 6 & S6*).

In revised manuscript, texts addressing main concerns are highlighted in blue.

---

### Meta-Review · Area_Chair_NZ2t · 2026-01-07

**Summary:**

This work identifies why multi‑task LoRA collapses when the number of tasks grows and proposes mtLoRA, which combines  (1) spectral‑aware regularization that orthogonalizes low‑singular‑value components while preserving high‑SV shared knowledge, (2) block‑level adaptation that attaches LoRA at the Transformer‑block level to cut gradient conflict and halve parameter count, and (3) fine‑grained routing that uses dimension‑specific weights instead of a single scalar. Empirically the method shows large gains (≈2 %–3 % over HydraLoRA) on vision and language benchmarks and the ablations demonstrate that each component contributes positively.

**Reviewer Concerns:**

- Reviewer M561 criticize the limited SOTA baseline set (mainly HydraLoRA) and request broader comparisons (MoLE, TIES‑Merging, DARE).
- Reviewers sPqK and fc95 view the contribution as incremental because each module exists in prior work, whereas Reviewer PCr8 describes the spectral analysis and integration as innovative and of significant impact.
- Reviewer M561 points out contradictory descriptions of $g$, while Reviewer PCr8 also raises a more general concern about the depth of routing‑granularity ablations.
- Cost: Reviewer M561 highlights the un‑benchmarked cost of per‑epoch SVD and the lack of justification for using the SVD of B as a proxy for the full update Δ; Reviewer PCr8 and the others do not emphasize this issue (PCr8 mentions routing overhead but not SVD).
- Scalability: Reviewer PCr8 questions the modest scale of experiments (max ≈ 25 tasks) and the lack of tests on very large LLMs, whereas Reviewer M561 and Reviewer sPqK consider the presented results (up to 25–100 tasks) sufficient to claim scalability.
- Reviewer sPqK stresses that orthogonal regularization, routing, and adapter placement are known, while Reviewer PCr8 focuses on the combination and its impact, leading to differing views on the paper’s overall originality.

**Reviewer Scores:**

- All reviewers acknowledge that the paper presents strong empirical results, achieving high accuracy on both vision (DOTA, iNat2018) and language (Dolly‑15k, BBH) benchmarks and that mtLoRA consistently outperforms the previous state‑of‑the‑art (HydraLoRA) by roughly 2 %–3 % average. These were extended and improved during rebuttal.

- Reviewers agreed that the three proposed components (spectral‑aware regularization, block‑level adaptation, and fine‑grained routing) are well‑integrated, supported by ablation studies, and that each component contributes positively to performance and gradient‑conflict reduction.

- All reviewers noted that reproducibility was initially weak (broken code link) but that the authors have now provided a working archive and additional details (standard deviations, extra visualizations).

- In response to reviewer's sPqK main concern, the authors stated that spectral‑aware regularization can be viewed as an implicit representation‑alignment technique: it orthogonalizes low‑SV “noise” while preserving the high‑SV subspace that is shared across tasks, thereby giving an intuitive (though not formal) justification for the singular‑value‑based treatment. I think this only partially addressed the concern of "lack of theoretical justification".

- The reviewers liked that the modules are plug‑and‑play and that the method shows cross‑domain generality (vision Transformers and large language models).

- I feel the rebuttal has clarified the paper and contributions.

---

### Decision · Program_Chairs · 2026-01-26

Accept (Poster)